# Chemical inhibition of stomatal differentiation by perturbation of the master-regulatory bHLH heterodimer via an ACT-Like domain

Ayami Nakagawa [1,8], Krishna Mohan Sepuru [2,3,8], Shu Jan Yip[1,8], Hyemin Seo[2,3], Calvin M. Coffin[2,3], Kota Hashimoto[4], Zixuan Li[4], Yasutomo Segawa [5], Rie Iwasaki[1], Hiroe Kato[1], Daisuke Kurihara [1,6], Yusuke Aihara[1,7], Stephanie Kim[3], Toshinori Kinoshita [1], Kenichiro Itami [1], Soon-Ki Han[1,6], Kei Murakami [1,4,7] ✉ & Keiko U. Torii [1,2,3] ✉

Selective perturbation of protein interactions with chemical compounds enables dissection and control of developmental processes. Differentiation of stomata, cellular valves vital for plant growth and survival, is specified by the basic-helix-loop-helix (bHLH) heterodimers. Harnessing a new amination reaction, we here report a synthesis, derivatization, target identification, and mode of action of an atypical doubly-sulfonylated imidazolone, Stomidazolone, which triggers stomatal stem cell arrest. Our forward chemical genetics followed by biophysical analyses elucidates that Stomidazolone directly binds to the C-terminal ACT-Like (ACTL) domain of MUTE, a master regulator of stomatal differentiation, and perturbs its heterodimerization with a partner bHLH, SCREAM in vitro and in plant cells. On the other hand, Stomidazolone analogs that are biologically inactive do not bind to MUTE or disrupt the SCREAM-MUTE heterodimers. Guided by structural docking modeling, we rationally design MUTE with reduced Stomidazolone binding. These engineered MUTE proteins are fully functional and confer Stomidazolone resistance in vivo. Our study identifies doubly-sulfonylated imidazolone as a direct inhibitor of the stomatal master regulator, further expanding the chemical space for perturbing bHLH-ACTL proteins to manipulate plant development.

Plant growth and survival rely on stomata - adjustable valves on the shoot epidermis of land plants facilitating efficient gas exchange while minimizing water loss. Each stoma consists of paired guard cells that can adjust the aperture in response to the environment, such as light, drought, and $CO_2$ levels[1]. A wealth of studies has elucidated the molecular basis of signaling mechanisms leading to stomatal opening and closure[2,3]. In addition to stomatal movement, stomatal development is also influenced by environmental conditions and in turn

[1]Institute of Transformative Bio-Molecules (WPI-ITbM), Nagoya University, Nagoya, Aichi, Japan. [2]Howard Hughes Medical Institute, The University of Texas at Austin, Austin, TX, USA. [3]Department of Molecular Biosciences, The University of Texas at Austin, Austin, TX, USA. [4]Department of Chemistry, Kwansei Gakuin University, Sanda, Hyogo, Japan. [5]Institute for Molecular Science and SOKENDAI, Myodaiji, Okazaki, Japan. [6]Institute for Advanced Research (IAR), Nagoya University, Nagoya, Aichi, Japan. [7]PRESTO, Japan Science and Technology Agency (JST), Chiyoda, Tokyo, Japan. [8]These authors contributed equally: Ayami Nakagawa, Krishna Mohan Sepuru, Shu Jan Yip. ✉e-mail: kei.murakami@kwansei.ac.jp; ktorii@utexas.edu

impacts plants' water-use efficiency[4,5]. During leaf development, stomata are generated through a stereotypical sequence of cell division and differentiation events[6,7]. In dicotyledonous plants, including Arabidopsis, the stomatal-cell lineage is initiated via de novo asymmetric cell divisions in the protoderm, which create meristemoids, proliferating stomatal precursor cells with a stem-cell-like property. After a few rounds of asymmetric divisions, the meristemoid differentiates into a guard mother cell (GMC). The GMC then undergoes a single symmetric division to terminally differentiate a stoma[8].

The cell-state transitional steps during stomatal development are governed by the sequential actions of basic-helix-loop-helix (bHLH) transcription factors (TFs), SPEECHLESS (SPCH), MUTE, and FAMA, which form heterodimers with their partner bHLHs, SCREAM (SCRM, also known as ICE1) and SCRM2[9–12]. The activities of these stomatal bHLH modules are enforced by the inhibitory peptide-receptor kinase signaling pathways. Briefly, secreted peptide ligands EPIDERMAL PATTERNING FACTORs (EPFs) are perceived by their receptors, ERECTA-family receptor kinases and TOO MANY MOUTHS (TMM)[13–18]. This activates a downstream Mitogen Activated Protein Kinase (MAPK) cascade, which in turn inhibits the bHLH heterodimers via direct phosphorylation and degradation events[19–21].

Interestingly, these stomatal bHLH TFs possess a C-terminal ACT-Like (ACTL) domain, a small structurally defined module resembling the ACT domain found in diverse metabolic enzymes in bacteria and eukaryotes (named after aspartate kinase-chorismate mutase-tyrA prephenate dehydrogenase)[21,22]. Recently, we revealed that the SCRM ACTL domain serves as a heterodimer selectivity interface with SPCH, MUTE, and FAMA[23]. This rather surprising finding offers a potential for the ACTL domain of bHLH TFs as a drug target for selective inhibition of heterodimerization, hence stomatal differentiation. Whereas the ACTL domain is predominant amongst plant bHLH proteins[24,25] no chemical inhibitors or binders targeting this domain, or more broadly, bHLH-ACTL TFs are available.

Chemical compounds that selectively perturb functions and interactions of key regulatory proteins can be a powerful tool to dissect and manipulate the development and physiology of cells to whole organisms[26,27]. Stomata are no exception[28]. For example, previous chemical screens identified compounds that suppress stomatal movement[29,30]. Likewise, small compounds have been utilized to examine the roles of signaling kinases and phosphatases in stomatal development[31–33]. Chemical library screens identified bubblin, which severely disrupts proper cell polarity in dividing meristemoids[34], as well as those that increase the number of stomata[35]. However, in all the above cases, the chemical compounds impact multiple processes, thus causing pleiotropic effects and/or their targets remain unidentified.

In an effort to expand the chemical space for controlling stomatal function and development, we have created a collection of nitrogen-containing chemical compounds by utilizing a newly developed amination reaction[36]. Nitrogen-containing molecules are widely recognized for their potent biological activities[37]. Previous chemical screens and subsequent structure-activity-relationship studies have shown that sulfonamidated compounds SIM1 (Stomata Influencing Molecule) and its derivative SIM3* inhibit stomatal movement[38]. Here, we report a complex doubly-sulfonylated imidazolone compound named Stomidazolone as a potent inhibitor of stomatal differentiation. We discovered that Stomidazolone tightly binds to the MUTE ACTL domain and perturbs the SCRM-MUTE heterodimerization. Additionally, we synthesized and identified biologically-inactive Stomidazolone analogs that severely compromise MUTE binding and its potency to interfere with SCRM-MUTE. Finally, we rationally designed Stomidazolone-resistant versions of MUTE and demonstrated their efficacy in vivo. Together with our chemical synthesis pipelines, our work will expand the chemical and pharmacological space for bioactive compounds and open the future directions for selective perturbation of prominent ACTL-domain-containing plant bHLH TFs for transient manipulation of plant growth and development.

## Results

### Doubly-sulfonylated imidazolone inhibits stomatal differentiation

In our initial attempt to derivatize SIM1 (compound **1**)[38], we synthesized SIM1* (compound **2**) from 2,4-diphenylimidazole through C−H imidation followed by monodesulfonylation, and serendipitously found the first one-shot synthesis of doubly-sulfonylated imidazolone (Fig. 1a; See "Methods"). Treatment of 2,4-diphenyloxazole with chloramine B in the presence of CuI afforded this doubly-sulfonylated AYSJ929 (compound **3**, later named as Stomidazolone: Fig. 1a, Supplementary Data 1 for synthesis and NMR spectra of the synthesized compounds). Further optimization produced Stomidazolone in 42% yield. Our reaction enabled the conversion of diphenyloxazole into substituted imidazolone with two equivalents of chloramine (see Supplementary Data 1). Even after searching the Reaxys database, we failed to uncover any records of a reaction that transforming diphenyloxazole into imidazolones. The structure was unambiguously confirmed by X-ray crystallographic analysis, which clearly shows the acute angular shape of this doubly-sulfonylated imidazolone (Fig. 1b, Supplementary Table 1, Supplementary Data 1). Unlike SIM1 and its derivative SIM3*[38], this compound did not exhibit any discernable effects on stomatal movement using our standardized assays (Supplementary Fig. 1).

The synthesized compound features numerous aryl groups, oxygen and nitrogen atoms as hydrogen bond acceptors, as well as hydrogen atoms serving as hydrogen bond donors (see Fig. 1b; Supplementary Data 1), suggestive of potential bioactivity. We thus subjected it to our chemical library screening pipeline for stomatal development[35]. Intriguingly, its application to Arabidopsis seedlings conferred reduced number of stomata but concomitant increase in the number of meristemoids (Fig. 1c). These increased meristemoids properly express both early and late meristemoid reporter markers, *TMMpro::GUS-GFP* and *MUTEpro::nucYFP*, respectively (Fig. 1d). Further quantitative analysis of stomata and meristemoid densities confirmed the dose-dependency of the compound, now named Stomidazolone, to decrease the number of stomata (24.45%, 59.50%, and 69.53% decrease from mock to 20, 50, and 100 μM Stomidazolone, respectively) while dramatically increasing the number of meristemoids (315.8%, 801.5%, and 647.6% increase from mock to 20, 50, and 100 μM Stomidazolone, respectively) (Fig. 1e). On the other hand, total number of stomata and meristemoids are only slightly reduced by the Stomidazolone treatment (14.46% and 25.03% at 20 and 50 μM) (Fig. 1e). These results suggest that Stomidazolone inhibits stomatal differentiation at the transition from proliferating meristemoids to differentiation.

Our bioassay pipelines utilize a liquid culture system[35], which may impose some limitation to the utility of Stomidazolone. We therefore investigated whether Stomidazolone supplemented to solid media can be taken up to influence stomatal differentiation. Like in liquid culture, Stomidazolone in the solid media inhibited stomatal differentiation in a dose-dependent manner, with concomitant increase in arrested meristemoids (Supplementary Fig. 2a-c). These results suggest that Stomidazolone can be taken up by the seedling roots to impact stomatal development.

Stomidazolone does not affect overall seedling growth including root elongation and cotyledon expansion, even at 100 μM in liquid culture (Fig. 1f). Likewise, Stomidazolone in the solid media does not impact seedling morphology at 20 and 50 μM (Supplementary Fig. 2d). It is thus highly unlikely that the observed inhibition of meristemoid differentiation is due to general seedling growth retardation. We therefore conclude that Stomidazolone potently triggers the

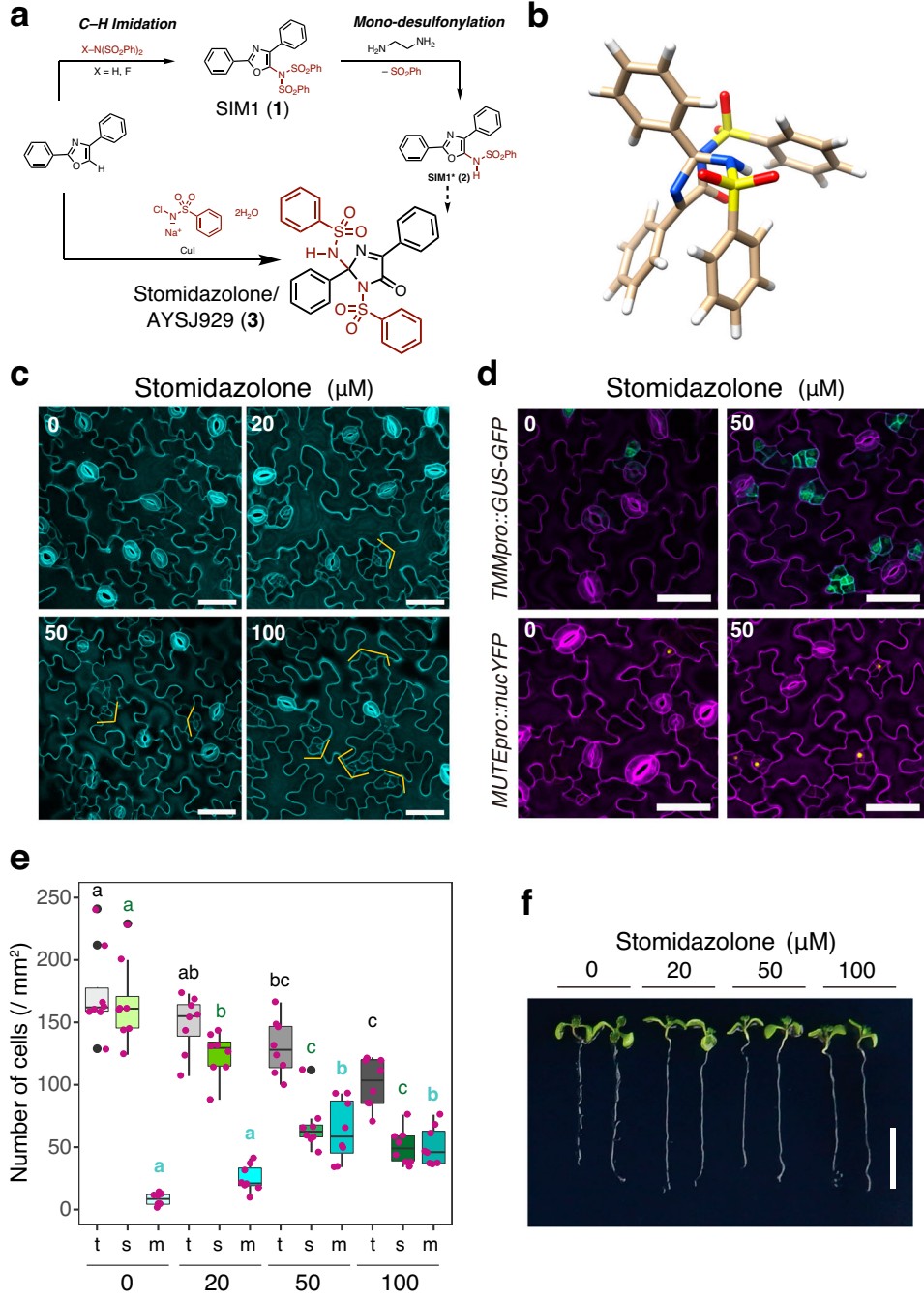

**Fig. 1 | Stomidazolone is a potent chemical for meristemoid arrest. a** Schematics of chemical synthesis. Supplementary Data 1 for the details of the synthesis. Bottom right, structure of Stomidazolone (ID: AYSJ929, compound **3**). **b** X-ray crystal structure of Stomidazolone shown as a stick model (White, H; Beige, C; Blue, N; Yellow, S; Red, O). **c** Confocal microscopy images of representative cotyledon abaxial epidermis from Arabidopsis wild-type (WT) seedlings 9-days-after germination (dag) grown in the presence of 0 (mock), 20, 50, and 100 μM Stomidazolone. Orange brackets, stomatal-lineage cells with an arrested meristemoid in the center. Scale bars, 50 μm. **d** Arrested meristemoids by Stomidazolone express proper meristemoid markers. Shown are confocal microscope images of 9-dag representative abaxial cotyledons epidermis expressing *TMMpro::GUS-GFP* (top) and *MUTEpro::nucYFP* (bottom) grown in the presence of 0 (mock: Left) and 50 μM Stomidazolone (Right). Scale bars, 50 μm (**e**) Stomidazolone treatment reduces the number of stomata but increases that of meristemoids. Quantitative analysis of stomatal (s; green), meristemoid (m; cyan), and stomata+meristemoid (t; gray) density per mm² of cotyledon abaxial epidermis from seedlings grown in the presence of 0 (mock), 20, 50, and 100 μM Stomidazolone. One-way ANOVA followed by Tukey's HSD analysis were performed for each cell type (stomata and meristemoids). Letters (**a**, **b**, **c**) indicate groups that are statistically different from other groups within each cell type. See "Methods" for the definition of the boxplots. See Source Data for the exact p values. $n = 8$ (**f**) Stomidazolone does not impact seedling growth. Shown are two representative 9-dag WT seedlings, grown in the presence of 0 (mock), 20, 50, and 100 μM Stomidazolone. Scale bars, 10 mm.

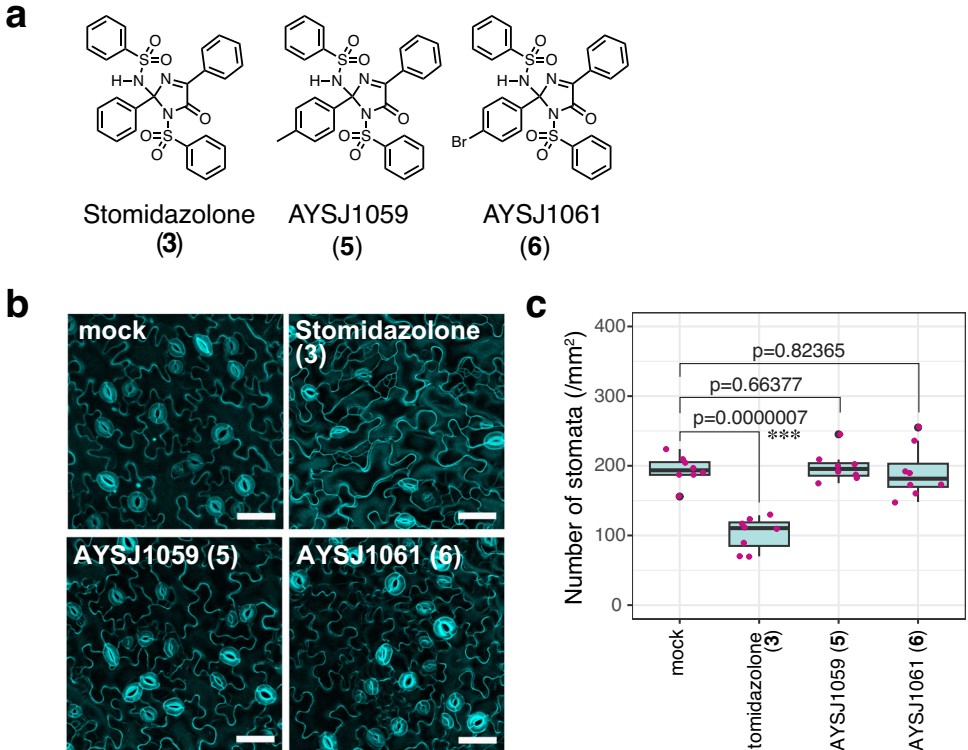

**Fig. 2 | Identification of biologically inactive Stomidazolone analogs.**
**a** Chemical structures of Stomidazolone (**3**) and its derivatives AYSJ1059 (**5**) and AYSJ1061 (**6**). See Supplementary Data 1 for detailed procedures of synthesis and analysis. **b** Representative confocal images of cotyledon abaxial epidermis from seedlings 9-days-after-germination grown with mock, or 50 μM Stomidazolone, or its derivatives grown on liquid culture medium. Scale bars, 50 μm. **c** Quantitative analysis. Number of stomata per 1 mm² of cotyledon abaxial epidermis from seedlings grown in the presence of mock or 50 μM Stomidazolone and its derivatives. Two-tailed Student's T test was performed, and *p* values are indicated. ***, *p* < 0.0005. See "Methods" for the definition of the boxplots. *n* = 8.

developmental arrest of the meristemoids, and in a lesser extent, inhibits the stomatal-lineage initiation.

## Stability of Stomidazolone and identification of inactive Stomidazolone analogs

To examine the stability of Stomidazolone and identify any degraded products (if unstable), we treated Stomidazolone with an acid or a base. As shown in Supplementary Data 1, we confirmed that Stomidazolone is stable under acidic conditions (with 3 equiv. of acetic acid) but completely degraded under basic conditions (with 3 equiv. of sodium hydroxide). The major degraded product under basic conditions was isolated with gel-permeation chromatography (GPC) in 48% yield. The product, which we designated as lzx103 (compound **4**), was identified as (phenylsulfonyl)benzimidamide[39] through ¹H and ¹³C NMR analyses (Supplementary Fig. 3a, Supplementary Data 1).

Next, to directly address the stability of Stomidazolone in the aqueous condition we used for our bioassays, we performed liquid chromatography-mass spectrometry (LC/MS) (Supplementary Fig. 4). Stomidazolone was incubated in the 1/2 MS liquid medium (pH 5.7) at room temperature for 48 hours. Stomidazolone peak was detected at the predicted elution time (2.47 min) whereas no lzx103 peak (2.13 min) was detected (Supplementary Fig. 4a, c, d). Additionally, we tested Stomidazolone's stability in phosphate buffer (pH 7.0). Again, our LC/MS analysis did not detect the degradation of Stomidazolone (Supplementary Fig. 4b, c, d). Based on these results, we conclude that Stomidazolone is a stable compound under biological conditions.

Subsequently, we performed bioassays of lzx103 (**4**) on developing Arabidopsis seedlings. Unlike Stomidazolone, the application of 50 μM lzx103 (**4**) did not cause any effects on stomatal development (Supplementary Fig. 3b, c, d). Our results rule out the possibility that the inhibition of stomatal differentiation is attributable to the

degradation product rather than Stomidazolone itself. These results establish Stomidazolone (**3**) as a bioactive chemical compound triggering the developmental arrest of stomatal precursor meristemoid cells.

Stomidazolone possesses a complex angular structure not shared by its precursor SIM1 (**1**) or its degradation product lzx103 (**4**: see Fig. 1, Supplementary Fig. 3, Supplementary Data 1). To further investigate the structure-activity relationship of Stomidazolone as an inhibitor of stomatal differentiation, we next synthesized two Stomidazolone analogs (Supplementary Data 1). Specifically, the Stomidazolone analogs, designated as AYSJ1059 (compound **5**) and AYSJ1061 (compound **6**) were synthesized from methylphenyl-substituted oxazole and bromophenyl-substituted oxazole using the copper-catalyzed conditions (Fig. 2a, Supplementary Data 1). Unlike Stomidazolone, the application of AYSJ1059 (**5**) as well as AYSJ1061 (**6**) at 50 μM to Arabidopsis seedlings did not affect the number of stomata (Fig. 2b, c), indicating that these two Stomidazolone analogs do not possess bioactivity to inhibit stomatal differentiation. Our results imply that the additional substituent (either methyl or bromo moieties) may perturb Stomidazolone's ability to bind to its target(s).

## Mechanistic genetic analyses delineate Stomidazolone as a potential inhibitor of MUTE

To unravel which pathways of stomatal development are impacted by Stomidazolone, we conducted mechanistic studies leveraged by the availability of a series of mutants known to regulate stomatal development (Fig. 3a). SPCH, MUTE, and FAMA consecutively drive stomatal-lineage progression[9–11] (Fig. 3a). To address whether Stomidazolone interferes with each cell-transitional step specified by these stomatal bHLH proteins, we applied Stomidazolone to *spch*, *mute*, and *fama* mutant seedlings. As shown in Fig. 3b, c,

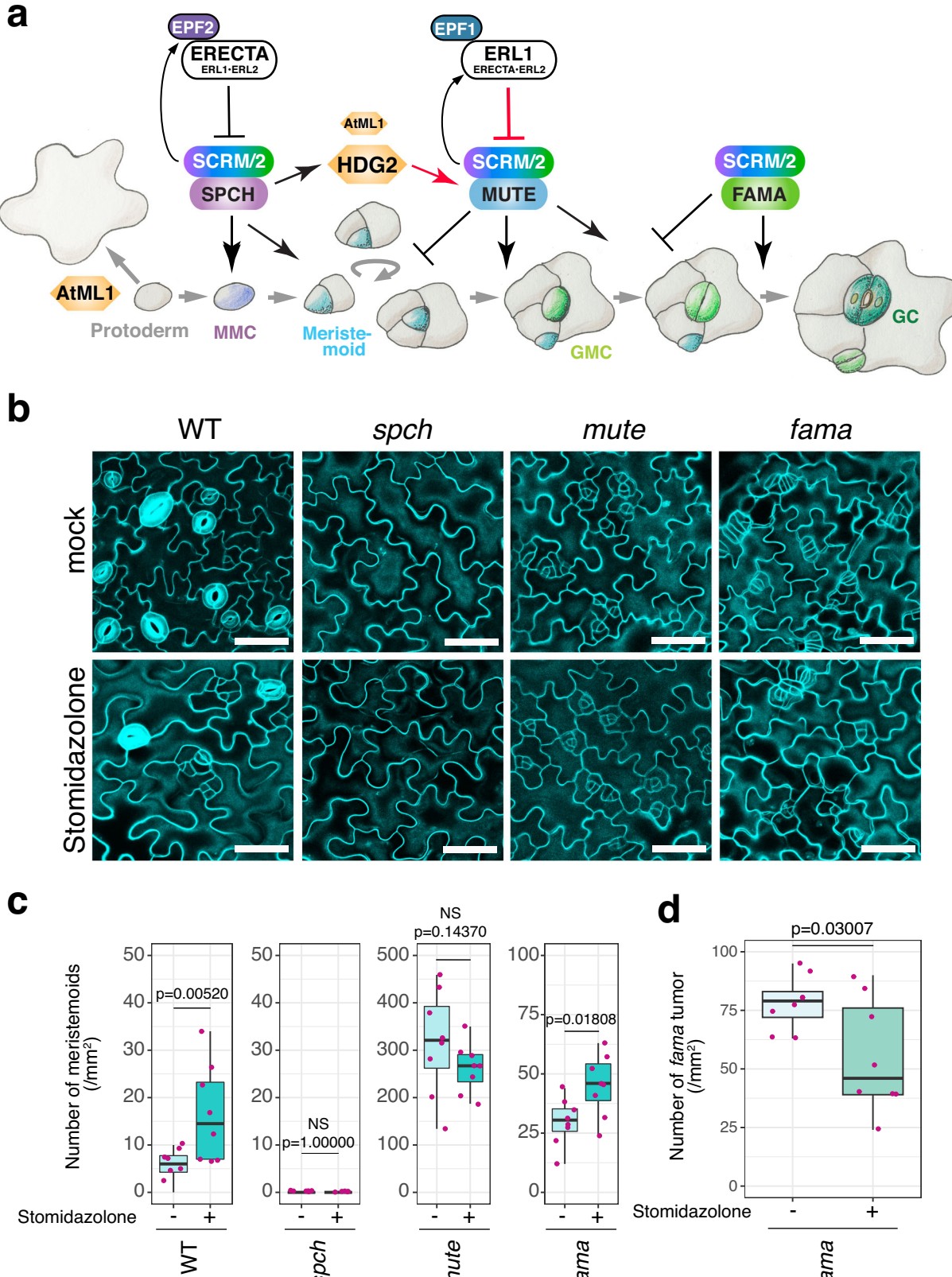

Stomidazolone exerted no effects on the pavement cell-only epidermis of *spch*. Likewise, the arrested meristemoid phenotype of *mute* was not enhanced by Stomidazolone. On the other hand, Stomidazolone significantly reduced the number of *fama* tumors whereas increasing the number of arrested meristemoids (Fig. 3c, d). Thus, *spch* and *mute* mutations are epistatic to

Stomidazolone, suggesting that this compound inhibits the MUTE-step of stomatal fate commitment.

It is established that *ERECTA*-family genes interact synergistically to enforce stomatal development[14]. Among them, *ERECTA* primarily inhibits the entry into stomatal cell lineages, a process regulated by *SPCH*, whereas *ERECTA-LIKE1* (*ERL1*) restricts the

**Fig. 3 | *spch* and *mute*, but not *fama*, are epistatic to Stomidazolone.**
**a** Schematic diagram showing the genetic control of stomatal cell-state transitions. Key genes/gene products and their roles are indicated in the cartoon. See main text for details of each step. Arrows, positive regulation; T-bars, negative regulation. Upregulation of *MUTE* by *HDG2* and inhibition of *MUTE* by *ERL1* are highlighted in red. MMC, meristemoid mother cell; GMC, guard mother cell; GC, guard cell. Cartoons modified from Han and Torii (2016)[6]. **b** Representative cotyledon abaxial

epidermis of mock and 50 μM Stomidazolone-treated WT, *spch (spch-3)*, *mute (mute-1)*, and *fama*. Scale bars, 50 μm. **c**, **d** Quantitative analysis of meristemoids density per 1 mm² (**c**) and *fama* tumors (**d**) of cotyledon abaxial epidermis from seedlings grown in the presence of 0 (mock) and 50 μM Stomidazolone. Two-tailed Student's T test was performed for each plant. p values and NS (not significant) are indicated above each boxplot. See "Methods" for the definition of the box-plots. *n* = 8.

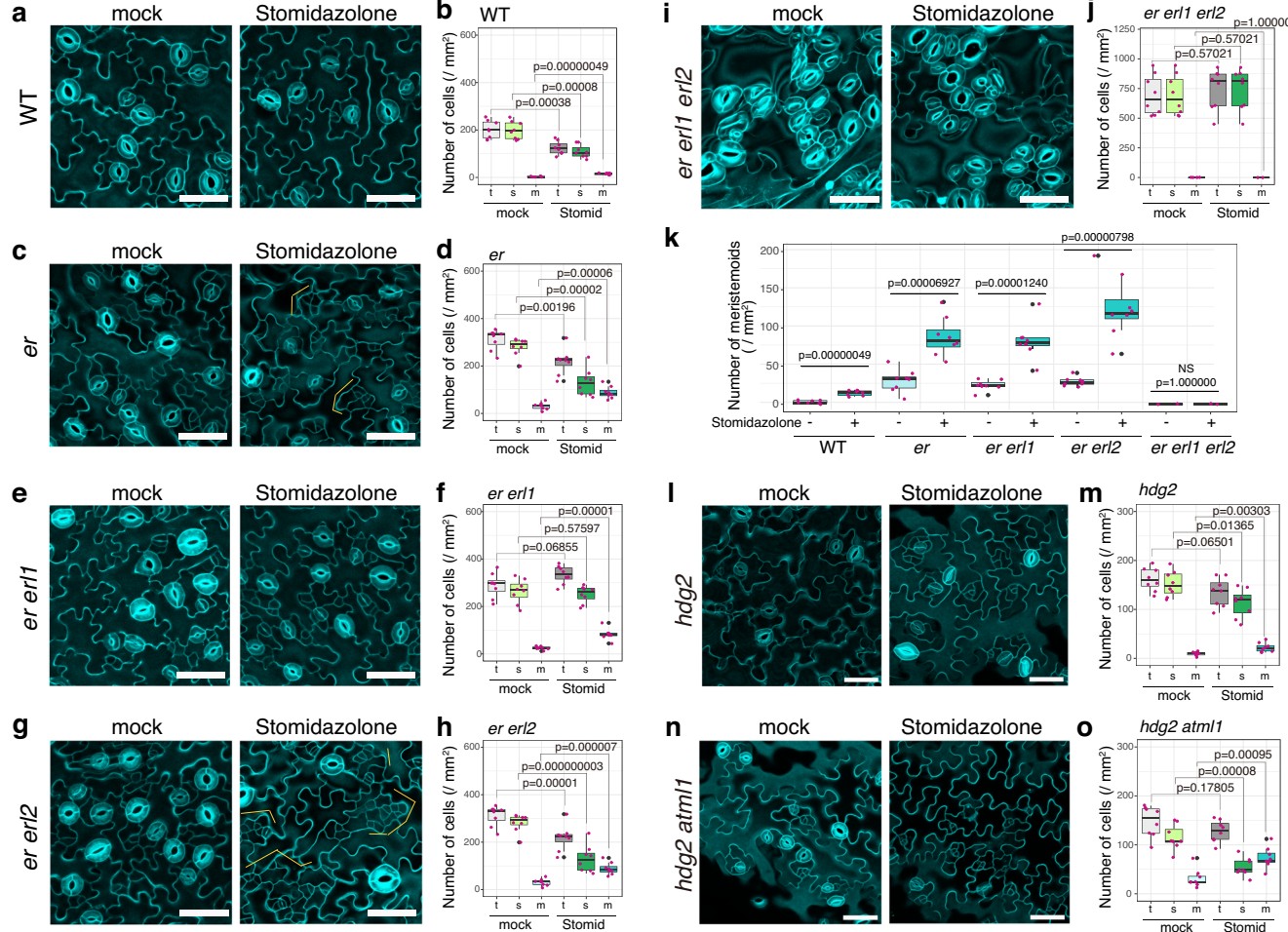

**Fig. 4 | Stomidazolone is a potent inhibitor of cell-state transition at the MUTE-step.** **a**–**j** Effects of Stomidazolone on the epidermis of wild-type (WT: **a**, **b**), *er* (**c**, **d**), *er erl1* (**e**, **f**), *er erl2* (**g**, **h**), and *er erl1 erl2* (**i**, **j**). Representative cotyledon abaxial epidermis of mock- (left) and 50 μM Stomidazolone (right). Orange brackets, clusters of arrested meristemoids. Scale bars, 50 μm. (**b**, **d**, **f**, **h**, **j**) Quantitative analysis of stomatal (s; green), meristemoid (m; cyan), and stomata+meristemoid (t; gray) density per mm² of cotyledon abaxial epidermis from seedlings grown in the presence of 0 (mock) and 50 μM Stomidazolone (Stomid). Two-tailed Student's

T test was performed for each plant. *p* values are indicated above each boxplot. See "Methods" for the definition of the boxplots. *n* = 8. **k** Number of meristemods, data replotted from other panels. Two-tailed Student's T test was performed for each plant. *p* values are indicated above each boxplot. See "Methods" for the definition of the boxplots. *n* = 8. **l**–**o** Effects of Stomidazolone on the epidermis of *hdg2* (*hdg2-2*: **l**, **m**) and *hdg2 atml1* (*hdg2-2 atml1-3*: **n**, **o**). All experiments and analyses were performed as described above. Scale bars, 50 μm. *n* = 8.

differentiation potential of the meristemoids, a process orchestrated by *MUTE*[40–42] (Fig. 3a). The loss of function in these receptor genes compromise the MAPK activation that inhibits SPCH and MUTE protein accumulation[19,20,41]. Compared to wild type, *erecta* null allele (*er*: *er-105*) overly produces stomatal-lineage cells[14] (Fig. 4a–d). Stomidazolone treatment exaggerated the *er-105* phenotype: the number of stomata drastically decreased and the meristemoids vastly increased (Fig. 4a–d, k). In *er erl1*, Stomidazolone treatment did not reduce the number of stomata, although it increased the number of meristemoids (Fig. 4e, f, k). The result supports the previous finding that the loss of *ERL1* intensifies stomatal differentiation by over-accumulation of MUTE[41], thereby

conferring partial resistance to Stomidazolone. The number of arrested meristemoids is even more increased in *er erl2* (Fig. 4g, h, k), consistent with the loss of *ERL1* in this genetic background. Finally, *er erl1 erl2* triple mutant seedlings appeared insensitive to Stomidazolone (Fig. 4i, j). Combined, these rather complex results further support that Stomidazolone primarily interferes with the MUTE activity for commitment to stomatal differentiation. Our results also suggest that if ERECTA-family mediated signaling pathways are absent, Stomidazolone cannot effectively interfere with the overly activated stomatal differentiation pathway.

It is known that HD-ZIP IV-family gene *HDG2* positively regulates stomatal differentiation by promoting *MUTE* expression and that its

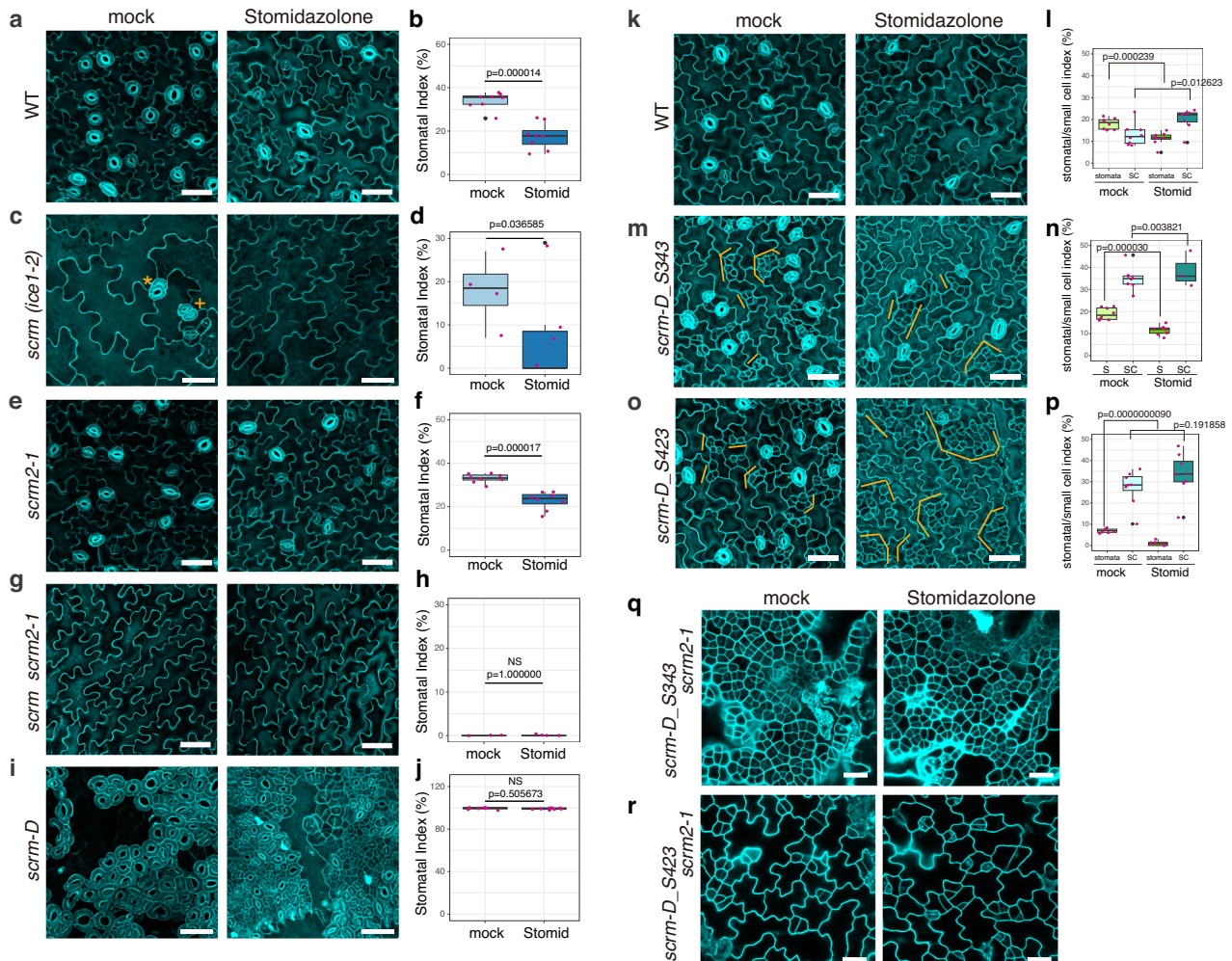

**Fig. 5 | SCRM activity dictates in vivo potency and selectivity of Stomidazolone.** **a**, **c**, **e**, **g**, **i** Abaxial epidermis of cotyledons in wild-type (WT: **a**), *scrm* (*ice1-2*: **c**), *scrm2-1* (**e**), *scrm scrm2-1* (**g**), and *scrm-D* (**e**) 9-dag seedlings treated with mock or 50 µM Stomidazolone. *scrm* loss-of-function mutants occasionally exhibit parallelly paired-stomata, so called 'four-lips' phenotype (orange plus) as well as stomata made with an extra guard cell (orange asterisk) likely due to *FAMA* mis-regulation. Scale bars, 50 µm. **b**, **d**, **f**, **h**, **j** Stomatal index (SI, %) for mock- or Stomidazolone-treated (Stomid) WT (**b**), *scrm* (**d**), *scrm2-1* (**f**), *scrm scrm2-1* (**h**), and *scrm-D* (**j**). See "Methods" for the definition of the boxplots. *n* = 8. (*n* = 7 for *scrm scrm2-1* (**h**)). Two-tailed Student's T-test was performed for pairwise comparison. **k**, **m**, **o** Abaxial epidermis of cotyledons in WT (**k**) and the intragenic *scrm-D* suppressors: *scrm-*

*D_S343* (**m**) and *scrm_D_S423* (**o**) 9-dag seedlings treated with mock or 50 µM Stomidazolone. Clustered meristemoids and their sister stomatal-lineage ground cells (SLGCs: orange brackets) become exaggerated. Scale bars, 50 µm. Experiments were repeated twice. **l**, **n**, **p** Stomatal and small cell (SC) index (%) for Mock- or Stomidazolone-treated (Stomid) WT (**l**), *scrm-D_S343* (**n**) and *scrm_D_S423* (**p**). See "Methods" for the definition of the boxplots. *n* = 8. Two-tailed Student's T-test was performed for pairwise comparisons. Experiments were repeated three times. **q**, **r** Abaxial epidermis of cotyledons in the intragenic *scrm-D* suppressors in the absence of *SCRM2*: *scrm-D_S343 scrm2-1* (**q**) and *scrm_D_S423 scrm2-1* (**r**) 9-dag seedlings treated with mock or 50 µM Stomidazolone. Experiments were repeated three times. Scale bars, 20 µm.

paralog *AtML1* enhances the *hdg2* mutant phenotype[43] (Fig. 3a). We next tested whether loss-of-function mutations in these genes enhance the effects of Stomidazolone to trigger meristemoid arrests. As shown in Fig. 4l, m, Stomidazolone increased the meristemoid numbers of *hdg2* epidermis while the total numbers of stomata and meristemoids were unaffected. Strikingly, *hdg2 atml1* double mutant exhibited hypersensitivity to Stomidazolone, and its application exaggerated *mute*-like epidermal phenotype (Fig. 4n, o). Taken together, a series of chemical-genetic dissections indicates that Stomidazolone likely inhibits the activity of MUTE.

### In vivo potency and selectivity of Stomidazolone relies on SCREAM

SCRM and its partially redundant paralog, SCRM2, serve as a heterodimeric partner of MUTE as well as SPCH and FAMA, and their activity is unequally required for stomatal cell-state transitions[12] (Fig. 3a). We investigated whether the activity of SCRMs influences the effects of

Stomidazolone on stomatal differentiation (Fig. 5). Stomidazolone treatment to *scrm* loss-of-function mutant (also known as *ice1-2*) severely reduced stomatal lineage initiation, resulting in a *spch*-like epidermis vastly composed of pavement cells (Fig. 5a–d). This result is different from other core stomatal development mutants, whereby Stomidazolone triggered a *mute*-like phenotype (Figs. 3, 4). Unlike *SCRM*, *SCRM2* knockout mutant *scrm2-1* does not confer any discernable phenotype[12]. The *scrm2-1* seedlings responded to Stomidazolone treatment just like wild-type seedlings: reduction of the number of stomata with meristemoid arrests (Fig. 5e, f). The complete loss of *SCRMs*, *scrm scrm2-1* double mutant, develops pavement cell-only epidermis identical to *spch*[12]. Stomidazolone has no effects on *scrm scrm2-1* epidermis (Fig. 5g, h). Conversely, a constitutively active form of SCRM, *scrm-D* confers stomata-only epidermis[12,21]. This *scrm-D* phenotype was not affected by Stomidazolone (Fig. 5i, j). These results suggest that Stomidazolone's potency as an inhibitor of stomatal differentiation relies on the activity of SCRM. In the absence of SCRM,

Stomidazolone can additionally inhibit the SPCH-regulated step of stomatal-lineage initiation. Here, SCRM2 is the only available partner for SPCH/MUTE/FAMA and as such, Stomidazolone might be able to target SCRM2 and SPCH.

Next, we sought to delineate if Stomidazolone interferes with the in vivo heterodimerization potential of SCRM. Recently, we have shown that the SCRM C-terminal ACTL domain serves as a hetero-dimerization interface and its dysfunctions selectively abrogate het-erodimerization of SCRM with MUTE, but not SPCH[23]. We therefore used our *scrm-D* intragenic suppressor mutants, *scrm-D_S343* and *scrm-D_S423*, which encode scrm-D$_{L484F}$ and scrm-D$_{\Delta ACTL}$, respectively[23]. As reported previously[23], both *scrm-D_S343* and *scrm-D_S423* exhibit nearly wild-type epidermis. Stomidazolone triggered excessive for-mation of arrested meristemoids in these *scrm-D* intragenic sup-pressors (Fig. 5k–p). The phenotype is slightly weaker, yet resembles to that of *scrm-D mute*[44], further supporting the hypothesis that Stomi-dazolone perturbs the activity of MUTE.

In the absence of *SCRM2*, both *scrm-D_S343* and *scrm-D_S423* confer massive clusters of arrested meristemoids, identical to the phenotype of *scrm-D mute*[23]. This occurs because scrm-D ACTL-domain mutant proteins (scrm-D$_{L484F}$ and scrm-D$_{\Delta ACTL}$) cannot stably heterodimerize with MUTE while still heterodimerizing with SPCH[23]. We thus predicted that if Stomidazolone mainly targets the SCRM-MUTE (but not SCRM-SPCH) heterodimer, its application will not affect the massive meristemoid cluster phenotype of *scrm-D_S343* and *scrm-D_S423* in the *scrm2* null mutant background. Consistent with this hypothesis, both *scrm-D_S343 scrm2-1* and *scrm-D_S423 scrm2-1* double mutant showed insensitivity to Stomidazolone (Fig. 5q, r). Taken together, our intricate in vivo phenotypic evidence highlights that Stomidazolone inhibits meristemoid differentiation, most likely by disrupting the heterodimerization of MUTE and SCRM through the C-terminal ACTL domain.

**Stomidazolone directly targets MUTE and interferes with its heterodimerization**

To address the molecular action of Stomidazolone, we investigated whether Stomidazolone directly disrupts the heterodimerization of SCRM with MUTE (as well as other stomatal-lineage bHLHs). First, we performed yeast two-hybrid (Y2H) assays. The growth assays show that Stomidazolone diminishes the association of SCRM with MUTE as well as with SPCH and FAMA (note that the N-terminal activation domain of SPCH and FAMA have been removed to pre-vent their autoactivation)[21] (Fig. 6a). Further quantitative α-galactosidase assays show that Stomidazolone inhibits these sto-matal bHLH heterodimers in a dose-dependent manner, with slightly less impact on the SPCH-SCRM heterodimers (Fig. 6b). Unlike Stomidazolone, its precursor analog SIM3* (compound **7**) has no effects on stomatal development nor heterodimerization of stomatal bHLH proteins (Supplementary Fig. 5).

To examine the effects of Stomidazolone on the hetero-dimerization of SCRM with full-length SPCH, MUTE, or FAMA in the context of the plant cells, we next performed ratiometric bimole-cular fluorescent complementation (BiFC) assays. Briefly, *Nicotiana benthamiana* leaves are transfected with a combination of SCRM and SPCH/MUTE/FAMA, each fused with a complementary half YFP, along with a full-length RFP (mScarlet-I3) fused with Histone H2B. Subsequently, leaf disks were prepared and treated with mock- or Stomidazolone (see Materials and Methods for details). In the absence of Stomidazolone, strong YFP signals were detected in the nuclei expressing the combination of SCRM-cYFP with SPCH/MUTE/FAMA-nYFP along with RFP signals (Fig. 6c). Strikingly, however, Stomidazolone treatment severely diminished the YFP signals in SCRM-cYFP + MUTE-nYFP pairwise combination (Fig. 6c). On the other hand, its effects on SCRM-cYFP + SPCH-nYFP or SCRM-cYFP + FAMA-nYFP were unclear (Fig. 6c). Subsequent quantitative

analyses of YFP/RFP signal ratio confirm that Stomidazolone severely perturbs the heterodimerization of SCRM-MUTE while weakly affecting SCRM-SPCH but not affecting SCRM-FAMA (Fig. 6d).

We observed that Stomidazolone can confer a weak *spch*-like phenotype when applied to *scrm* mutant seedlings (Fig. 5c, d). This rather unexpected phenotype can be explained if Stomidazolone effectively blocks the heterodimerization of SPCH with SCRM2. To further address this possibility, we conducted rBiFC assays with SCRM2 and SPCH/MUTE/FAMA. Like SCRM, SCRM2 hetero-dimerizes with SPCH, MUTE, and FAMA, as detected by YFP signal reconstitutions in the nuclei (Supplementary Fig. 6a, b). Unlike the case of SCRM, however, Stomidazolone treatment abolished the heterodimerization of SCRM2 with all of SPCH/MUTE/FAMA (Supplementary Fig. 6a, b). The findings that Stomidazolone dis-rupts SCRM2-SPCH, but not SCRM-SPCH heterodimers fully explain the observed in vivo effects on Stomidazolone on *scrm* mutant (Fig. 5c, d). Additionally, perturbation of SCRM2-MUTE hetero-dimers by Stomidazolone is consistent with the exaggerated mer-istemoid arrest phenotypes by Stomidazolone treatment on *scrm-D_S343* and *scrm-D_S423* seedlings (Fig. 5k–p).

To quantitatively assess the direct inhibition of SCRM-MUTE heterodimers by Stomidazolone, we further performed two in vitro biophysical protein-protein interaction kinetics assays using biolayer interferometry (BLI) and isothermal titration calorimetry (ITC) (Fig. 6e, f; Supplementary Figs. 7-9, Supplementary Tables 2, 3). For this purpose, recombinant full-length SCRM and MUTE proteins were expressed and purified. Preincubation of Stomidazolone and MUTE decreased the binding affinity of MUTE and SCRM by two orders of magnitude for both BLI and ITC methods (Kd values increased from $7.5 \pm 1.2$ nM to $856.5 \pm 15.1$ nM in BLI and from $9.5 \pm 1.6$ nM to $900 \pm 34.5$ nM in ITC for SCRM-MUTE binding without or with pre-incubation with Stomidazolone, respectively) (Fig. 6e, f, Supplemen-tary Fig. 7). Consistent with the lack of biological activity to inhibit stomatal differentiation, the Stomidazolone analogs AYSJ1059 (**5**) and AYSJ1061 (**6**) showed minimal effects on the SCRM-MUTE hetero-dimerization (Supplementary Fig. 9, Supplementary Tables 2, 3). These results demonstrate that Stomidazolone directly interferes with the SCRM-MUTE heterodimerization.

We next tested whether Stomidazolone directly binds to MUTE. Indeed, the BLI assay showed Stomidazolone binding to MUTE (Kd = $8.9 \pm 1.2$ μM) (Fig. 6g). Again, biologically inactive Stomidazolone analogs, AYSJ1059 (**5**) and AYSJ1061 (**6**) (Fig. 2), exhibit no binding to MUTE when analyzed by ITC and substantially weak binding by BLI (Supplementary Fig. 8, Supplementary Tables 2, 3). In addition, we found that Stomidazolone binds with recombinant SPCH, FAMA, and SCRM proteins, albeit with lower affinity (Fig. 6g, Supplementary Fig. 8a, b, d). These results support that inhibition of stomatal differ-entiation by Stomidazolone primarily occurs via its direct binding to MUTE.

Finally, to identify the binding domain of Stomidazolone on MUTE, we performed BLI assays with the purified recombinant bHLH domain and ACTL domain of MUTE. As shown in Fig. 6h, Stomidazo-lone specifically binds to the ACTL domain (Kd = $13.1 \pm 1.3$ μM). Based on the ITC data, the binding is primarily driven by enthalpic interac-tions (Supplementary Table 4). Again, Stomidazolone showed a lower affinity with the ACTL domains of SCRM, SPCH, and FAMA (Supple-mentary Fig. 8e, f, h, Supplementary Table 3). Together with a series of *in planta* and in vitro studies, the results indicate that Stomidazolone directly and primarily targets the ACTL domain of MUTE and interferes with the SCRM-MUTE heterodimerization. Furthermore, our results reinforce the previous finding that the ACTL domain integrity is strictly required for the heterodimerization of SCRM-MUTE[23], and convin-cingly provide the molecular, mechanistic basis of Stomidazolone as an inhibitor of meristemoid differentiation.

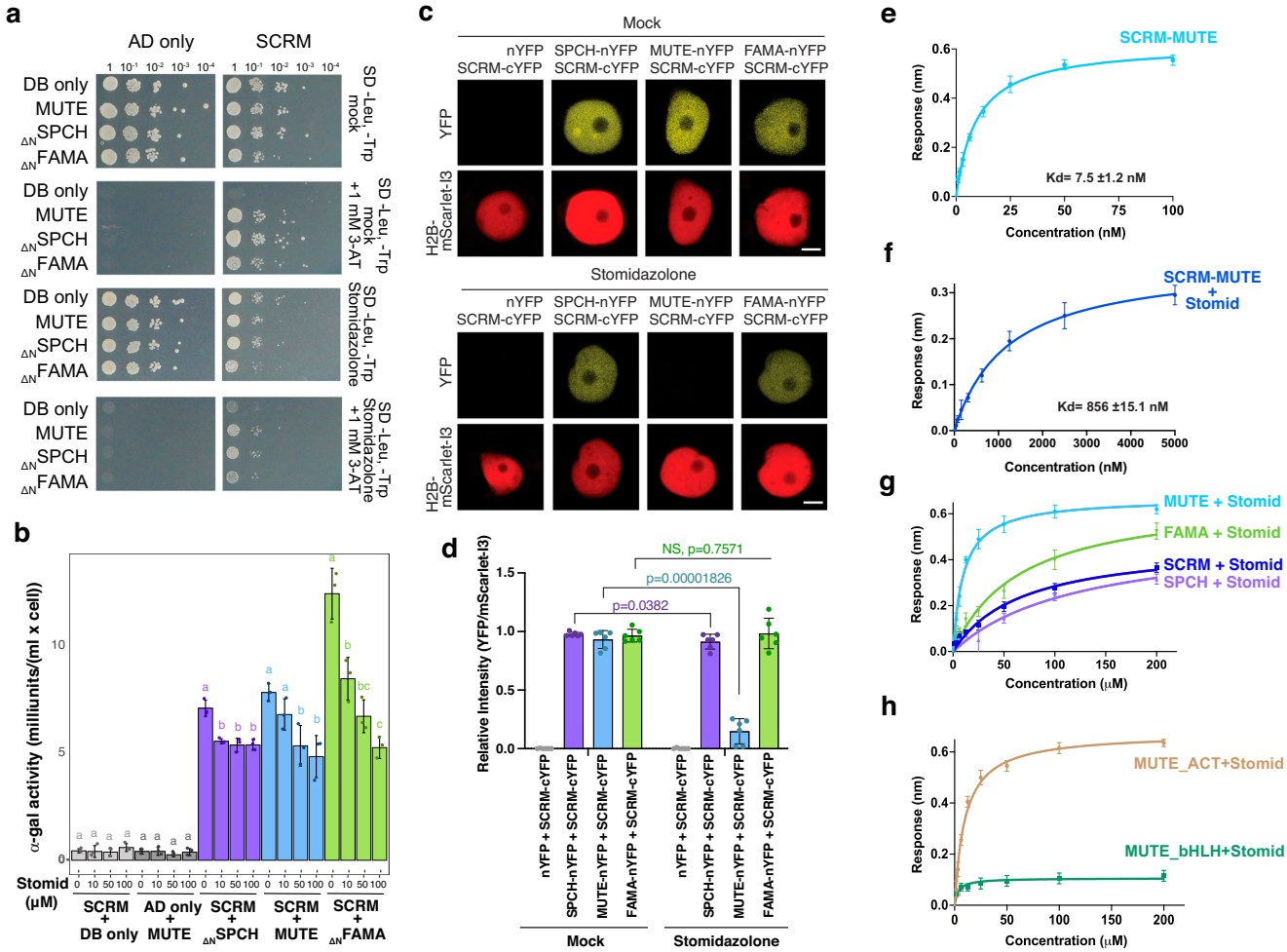

**Fig. 6 | Stomidazolone primarily disrupts SCRM-MUTE heterodimers through direct binding. a** Y2H analysis. Yeasts expressing control and pairs of stomatal bHLH proteins were spotted in 10-fold serial dilutions on appropriate selection media with or without 50 μM Stomidazolone. 1 mM 3-AT treatment shows that Stomidazolone reduces the interactions of SCRM with stomatal bHLH proteins. 3-AT, 3-amino-1,2,4-triazole. **b** Quantitative α-galactosidase assays. Yeasts expressing control and pairs of stomatal bHLH proteins were cultured in 0, 10, 50, and 100 μM Stomidazolone and enzyme assays were performed. Values indicate mean ± SD. One-way ANOVA followed by Tukey's HSD analysis were performed. Letters (a, b, c) indicate that the enzyme activities are statistically different within the group. See Source Data for the exact p values. Experiments were repeated 3 times. **c** Ratiometric BiFC analysis. *N. benthamiana* leaves were infiltrated with Histone H2B (H2B)-mScarlet-I3 and pairs of stomatal bHLH proteins. Subsequently, leaf disks were treated either with mock (top) or 100 μM Stomidazolone (bottom). Shown is a representative nucleus from each combination imaged simultaneously for mScarlet-I3 (control) and YFP (interaction) signals. Scale bars, 5 μm.

Experiments were repeated three times. Values indicate mean ± SD. **d** Quantitative analysis of BiFC YFP signal intensity ratio normalized by the signal intensity of H2B-mScarlet-I3. Two-tailed Student T-test was performed for each pairwise comparison. NS, not significant. *n* = 6. Experiments were repeated three times. **e**, **f** Quantitative analysis of SCRM-MUTE interactions in the absence (**e**), and presence (**f**) of Stomidazolone. In vitro binding response curves are provided for SCRM with MUTE and with/without Stomidazolone. The data represent the mean ± SD and are representative of two independent experiments. **g** Quantitative analysis of Stomidazolone binding with SCRM (blue), MUTE (cyan), FAMA (green), and SPCH (lavender). The MUTE protein exhibit higher affinity to Stomidazolone. In vitro binding response curves are provided Stomidazolone with SCRM/MUTE/SPCH/MUTE/FAMA. The data represent the mean ± SD and are representative of two independent experiments. **h** Quantitative analysis of Stomidazolone binding with MUTE bHLH (green) and ACTL (brown) domains. In vitro binding response curves are provided for Stomidazolone with MUTE$_{bHLH}$ and MUTE$_{ACTL}$. The data represent the mean ± SD and are representative of two independent experiments.

## Docking modeling identifies the Stomidazolone-MUTE binding interface

We demonstrated that the ACTL domain of MUTE is targeted by Stomidazolone (Fig. 6). To delineate the specific binding interface, we performed computational docking modeling (Supplementary Fig. 10). Briefly, we used HADDOCK-based calculations to drive the docking process, using mutagenesis data and binding site information from Swiss Dock as ambiguous interaction restraints. Structural models showed that different racemic mixtures of Stomidazolone adopted the same geometry in all clusters, with ionic interactions mediated by charged residues within the MUTE ACTL domain, Arg-133 (R133), Arg-134 (R134), Glu-162 (E162), and Glu-163 (E163), and H-bonding and packing interactions by polar and

hydrophobic residues Ile135, Val136, Thr164 and Val165 (Fig. 7a, b, Supplementary Fig. 10a). The binding free energies of both racemic mixtures were similar (Supplementary Fig. 10, Supplementary Table 4). In contrast, the biologically inactive Stomidazolone analogs AYSJ1059 (**5**) and AYSJ1061 (**6**), in both enantiomers, failed to converge into a discrete binding model (Supplementary Fig. 10b, c), consistent with its lack of direct binding to the MUTE ACTL domain detected with ITC (Supplementary Fig. 8l, m).

Amino-acid sequence alignments of MUTE and its orthologs as well as their sister bHLHs (SPCH and FAMA) show that R133 and R134 residues are highly conserved among the MUTE and FAMA clades, whereas substitutions to His (H) and Lys (K) are prevalent in the SPCH clade. The E162 and E163 residues are replaced by Glu (E) and

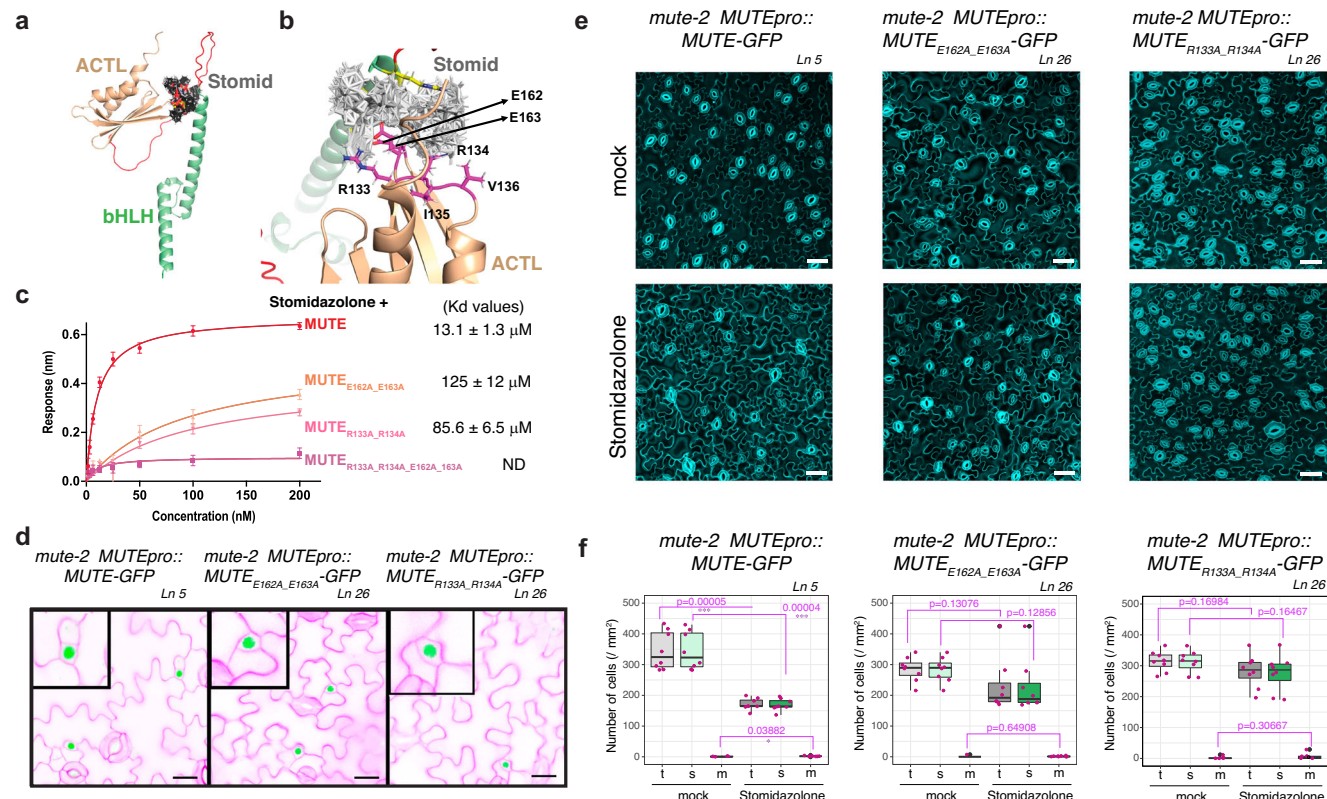

**Fig. 7 | Engineering Stomidazolone-resistant plants based on the MUTE-Stomidazolone docking model. a, b** MUTE-Stomidazolone binding models. **a** Alphafold2 predicted full-length MUTE protein is shown as ribbon representation. MUTE bHLH domain, green; ACTL domain, peach; Linker region, red. Stomidazolone is shown as stick model (gray). **b** Magnified view of MUTE-Stomidazolone binding interface. The key binding residues (R133, R134, E162 and E163) sidechains are highlighted in magenta. (**c**) Quantitative analysis of Stomidazolone binding to wild-type MUTE and MUTE site-directed mutants (MUTE$_{R133A\_R134A}$, MUTE$_{E162A\_E163A}$, MUTE$_{R133A\_R134A\_E162A\_E163A}$) by BLI. In vitro binding response curves are presented for Stomidazolone with MUTE and its mutants. Stomidazolone compound was tested at seven different concentrations (200, 100, 50, 25, 12.5, 6.25, and 3.125 μM). The data represent the mean ± SD and are representative of two independent experiments. **d** Site-directed mutant versions of MUTE that disrupt Stomidazolone binding are expressed normally and biologically functional. Representative cotyledon abaxial epidermis of 5-day-old *mute-2* seedlings expressing *MUTEpro::MUTE-GFP* (left), *MUTEpro::MUTE$_{E162A\_E163A}$ -GFP* (center), and *MUTEpro::MUTE$_{R133A\_R134A}$-* 

*GFP* (right). Insets, representative late meristemoids from each panel with MUTE-GFP signals in nucleus. Confocal images are inverted for clear cellular outlines. Observations were repeated at least 3 times, each with 8 to 10 seedlings per line. Scale bars, 20 μm. **e, f** MUTE mutant versions that disrupt Stomidazolone binding confer drug resistance in vivo. **e** Representative cotyledon abaxial epidermis of 9-day-old *mute-2* seedlings expressing *MUTEpro::MUTE-GFP* (left), *MUTEpro:: MUTE$_{E162A\_E163A}$ -GFP* (center), and *MUTEpro::MUTE$_{R133A\_R134A}$ -GFP* (right) with mock- (top) and 20 μM Stomidazolone for eight days (bottom). Scale bars, 50 μm. **f** Quantitative analysis of total (t: stomata and meristemoids), stomatal (s: green) or meristemoid (m: light cyan) density per mm² of cotyledon abaxial epidermis from seedlings grown in the presence of 0 (mock) and 20 μM Stomidazolone. Two-tailed Student's T test was performed were performed for each cell type (total, stomata and meristemoids) and *p* values are indicated. *, *p* < 0.05, ***, *p* < 0.0005. See "Methods" for the definition of the boxplots. Experiments were repeated twice. *n* = 8.

Gln (Q) and Asp (D) and D/E in FAMA and SPCH clades, respectively (Supplementary Fig. 11). Overall, the Stomidazolone-SPCH/MUTE/FAMA ACTL domain binding interfaces possess positively and negatively charged residues with clade-specific sequences, which might reflect the differences in physical binding affinities (Fig. 7b, Supplementary Fig. 11, Supplementary Tables 2, 3).

To confirm the predicted binding residues for Stomidazolone-MUTE interaction, we further performed Alanine substitutions. For this purpose, we generated MUTE$_{R133A\_R134A}$, MUTE$_{E162A\_E163A}$, and MUTE$_{R133A\_R134A\_E162A\_E163A}$ mutants and then quantitatively analyzed Stomidazolone binding affinity on these mutant versions of MUTE ACTL domain by using biophysical assays, BLI and ITC (Fig. 7c, Supplementary Fig. 8i, j, k and Supplementary Table 3). In both assays, the binding affinity was significantly reduced; The Kd values obtained for BLI assays are 85.6 ± 5.5 μM and 125.0 ± 12.0 μM for double mutants, and no binding was detected for the quadruple mutant (Fig. 7c, Supplementary Fig. 8i, j, k, Supplementary Tables 2, 3). These results provide the experimental evidence for the predicted Stomidazolone-binding interface of specific MUTE residues within its ACTL domain.

## Engineering Stomidazolone-resistance via a molecular docking modeling

Our molecular docking modeling and experimental verification deciphered the Stomidazolone binding interface at the MUTE ACTL domain (Fig. 7a, b). To the best of our knowledge, nothing is known about the bioactivity of doubly-sulfonylated imidazolones nor drugs that target the ACTL domains of bHLH TFs. We thus first explored the functional importance of the Stomidazolone-binding sites within the MUTE ACTL domain. We expressed two different *MUTE* mutant versions that are diminished in Stomidazolone binding (Fig. 7c) into *mute* knockout (*mute-2*) plants driven by its own promoter (*MUTEpro::MUTE$_{E162A\_E163A}$-GFP* and *MUTEpro:: MUTE$_{R133A\_R134A}$-GFP*). We in addition generated a control *MUTEpro::MUTE-GFP* construct using the identical molecular cloning strategy, which was also transformed into *mute-2* in parallel (see "Methods"). Like the wild-type MUTE-GFP, the engineered versions of MUTE-GFP are detected in the nuclei of a differentiating meristemoid, and their expression fully rescued the *mute* arrested meristemoid phenotype (Fig. 7d, e, Supplementary Fig. 12a). These transgenic *mute* null lines grow into

healthy, fertile plants indistinguishable from the wild type. Thus, the site-directed mutations within the predicted Stomidazolone-MUTE interaction interface do not impact the expression patterns or functionalities of MUTE.

Next, we tested the effects of Stomidazolone. As expected, control *MUTEpro::MUTE-GFP mute-2* seedlings significantly reduced the number of stomata upon Stomidazolone treatment (Fig. 7e, f, Supplementary Fig. 12b, c). In contrast, independent lines of *MUTEpro::MUTE$_{E162A\_E163A}$-GFP* and *MUTEpro:: MUTE$_{R133A\_R134A}$-GFP* exhibit Stomidazolone resistance, because Stomidazolone treatment did not significantly affect the numbers of stomata and meristemoids (Fig. 7e, f, Supplementary Fig. 12b, c). These results validate the structurally modelled Stomidazolone-MUTE binding interface and unambiguously demonstrate that the in vivo binding of Stomidazolone to the MUTE ACTL domain is the cause of the inhibition of stomatal differentiation. Furthermore, our structure-guided engineering of Stomidazolone-resistant plants suggests that the Stomidazolone-binding interface within the ACTL domain can be uncoupled from the biological activity of the bHLH-ACTL TFs.

## Discussion

In this study, we developed previously undescribed catalytic amination reactions and materialized the synthesis of Stomidazolone, a complex, doubly-sulfonylated imidazolone that selectively inhibits stomatal differentiation. Leveraged by the wealth of stomatal development mutant resources, our forward chemical genetics pinpointed the exact in vivo target of Stomidazolone: the master regulator MUTE. Further biochemical, biophysical, and structural modeling studies elucidated that Stomidazolone primarily perturbs the SCRM-MUTE heterodimerization through interacting with the ACTL-domain of MUTE. Chemical synthesis of TF dimerization inhibitors is an under/unexplored field in plant biology. As such, our work may expand a chemical space for manipulation of developmental processes.

In yeast, Stomidazolone interferes with the heterodimerization of not only SCRM-MUTE but also SCRM-SPCH and SCRM-FAMA (Fig. 6a, b). Remarkably, in the context of in vivo plant cells, however, Stomidazolone exhibits selective disruption of the SCRM-MUTE heterodimers (Fig. 6c, d) and, consistently, confers meristemoids arrest in developing seedling epidermis (Figs. 1, 3). It is known that SPCH, MUTE, and FAMA are closely related paralogs. Yet, they possess distinct domains, with MUTE being the smallest without the extended N-terminal domain[11]. SPCH, MUTE, and FAMA are known to associate with different protein complexes, including those mediating transcription and epigenetic regulations[45–47]. It is therefore plausible that association with distinct protein complexes may contribute to the in vivo selectivity of Stomidazolone toward SCRM-MUTE, which cannot be recapitulated in the heterologous yeast system (note that truncated SPCH and FAMA proteins lacking their N-terminal domain were used in yeast in order to avoid transcriptional auto-activation) (Fig. 6a, b). Moreover, prevalent effects of Stomidazolone to perturb SCRM-MUTE heterodimers may also reflect the inherent property of bHLH-ACTL heterodimerization. It has been shown that heterodimerization of SCRM-MUTE, but not SCRM-SPCH or SCRM-FAMA, relies on the presence or integrity of the SCRM ACTL domain[23]. Thus, our findings support the notion that the SCRM-MUTE pair is the most labile, hence druggable, amongst the stomatal bHLH heterodimers.

The in vitro Stomidazolone binding assays show that Stomidazolone most strongly associates with the ACTL domain of MUTE but also weakly associates with that of SPCH, FAMA, and SCRM (Fig. 6). Thus, whereas the specificity for perturbing the bHLH-ACTL heterodimerization may be impacted by the in vivo protein complex status, the specificity of direct binding of Stomidazolone to the ACTL domain lies in local protein sequence. In support of this idea, the key Stomidazolone-interacting amino-acid residues within the MUTE ACTL domain are not fully conserved in SPCH, FAMA, and SCRM

(Supplementary Fig. 11). Future structural analysis of Stomidazolone-ACTL binding interface may clarify these points.

Our exhaustive mechanistic genetic analyses with Stomidazolone revealed that, in the absence of SCRM, Stomidazolone vastly reduces the number of stomata, thus phenocopying the weak *spch* mutant[9] (Fig. 5c, d). This unexpected phenotype underscores the unequal redundancy between *SCRM* and its lesser-biologically-potent paralog *SCRM2*[12]. In this study, we showed that SCRM2-SPCH heterodimers, but not SCRM-SPCH heterodimers, are targeted in vivo by Stomidazolone (Supplementary Fig. 6). The high efficacy of Stomidazolone on perturbing SCRM2-SPCH/MUTE/FAMA heterodimerization may likely stem from the less stable, labile nature of SCRM2, which may result in weaker heterodimerization affinity. Consistently, it has been shown that the constitutively active version of SCRM2, SCRM2$_{R203H}$, which possesses the Arginine-to-Histidine substitution within its KRAAM motif identical to scrm-D, does not confer excessive stomatal differentiation phenotype as severe as *scrm-D*[12]. In any event, the presence of SCRM, including those with ACTL-domain mutations, can mask the effects of Stomidazolone on SCRM2 (Fig. 5). In contrast, the mutations that stabilize SCRM confer Stomidazolone insensitivity (Figs. 4i–k, 5i, j, q, r). They include loss of upstream receptor kinases (*er erl1 erl2*)[14], constitutively active *scrm-D*[12,21], as well as *scrm-D* ACTL domain mutants in the absence of *SCRM2*[23]. Because these mutations prevent SPCH from undergoing MAPK-mediated, phosphorylation-dependent degradation[19,21,23], the observed Stomidazolone insensitivity can be explained if these stabilized versions of SCRM-SPCH heterodimers can compensate for the reduction of SCRM-MUTE.

The discovery that Stomidazolone targets functionally dispensable residues within the MUTE ACTL domain brings several implications. First, Stomidazolone (and its derivatives) could be optimized in the future to map the functional significance of ACTL domains among the bHLH TFs. The ACTL domain-containing bHLH proteins are prominent within the land plant lineages[24]. Whereas studies indicated the roles of the ACTL domains in dimerization[25], their functional significance or mode of action remains largely unclear. In the well-studied case of Maize R, its ACTL domain functions as an auto-inhibitory homodimerization module that prevents the DNA binding[48,49], a mechanism different from the stomatal bHLH-ACTL proteins[23]. Second, the finding hints at a hitherto unidentified small ligand-binding property of the ACTL-domain-containing bHLH proteins. The prototypical ACT domain is found in biosynthetic enzymes and functions as a sensor domain for small metabolic ligands, including amino acids and thiamine[22]. On the contrary, it is unknown whether bHLH-ACTL TFs bind any endogenous small molecules. If they do, how Stomidazolone interferes with such endogenous ligands would be an interesting future direction. Third, as we successfully achieved for MUTE (Fig. 7, Supplementary Fig. 12), other bHLH-ACTL TFs could be engineered to generate drug-resistant versions to selectively modulate plant phenotypes without compromising their biological functions. From a practical point of view, Stomidazolone can be applied to both liquid and solid culture media to target specific bHLH-ACTL heterodimers and it is quite stable in the MS media (Supplementary Figs. 2, 4). Moreover, our streamlined synthesis of Stomidazolone analogs (Fig. 2, Supplementary Data 1) may be expanded to generate a library of potential ACTL-targeting imidazolones. In any event, further structural analyses would help elucidate the exact biological significance and applicability of Stomidazolone to delineate and manipulate bHLH-ACTL proteins.

Harnessing synthetic chemistry to disrupt TF dimerization - while it seems like an obvious means of manipulating growth and development, this approach is surprisingly unexplored in plants. Studies identified small proteins that can competitively interfere with TF dimers and prevent their DNA binding[50]. For instance, LONG HYPO-COTYL IN FAR-RED1 (HFR1) protein titrates the bHLH TF,

PHYTOCHROME INTERACTING FACTORs (PIFs), from forming functional dimers[51]. In mammalian systems, natural compound library screens have identified small molecules that disrupt the homo-dimerization of Hes1, an bHLH TF regulating neural cell proliferation and differentiation[52,53]. In all these cases, unlike Stomidazolone, the inhibitors target the core dimerization HLH modules that are essential for the TF function, thereby leaving little room for designing and engineering the drug resistant versions of said TFs.

In the synthetic chemistry point of view, the serendipitous discoveries of Stomidazolone and subsequent chemical synthesis proffered a catalytic transformation of oxazoles[37,54]. Our newly discovered reaction can convert diphenyloxazole into imidazolone to create highly angular compounds (see Supplementary Data 1), which are unexplored in a pharmacology field. Now that we demonstrated the interaction of this high degree of three-dimensionality of Stomidazolone interacts with a discrete structural module of the ACTL domain and selectively inhibits the key step of stomatal differentiation, further elaboration of our amination reaction may expand the chemical space for manipulating protein functions in plants and beyond.

## Methods

### Plant growth condition and chemical treatment

*Arabidopsis* accession Columbia (Col-0) was used as wild-type. The following mutants/transgenic lines are reported elsewhere: *TMMpro::GUS-GFP*[13], *spch-3/+, mute/+,* and *fama-1/+*[11]; *MUTEpro::nucYFP*[41]; *ice1-2/scrm* (SALK_003155), *scrm2-1* (SAIL_808_B10), and *ice1-2/scrm scrm2-1*[12]; *er-105 erl1-2 erl2-1*[14]; *hdg2-2* (SALK_127828C) and *hdg2-2 atml1-3* (SALK_128172)[43]; *scrm-D_S343, scrm-D_S423, scrm-D_S343 scrm2-1,* and *scrm-D_S423 scrm2-1*[23]. Their genotype and transcript reduction were confirmed. The presence of transgenes/mutant alleles were confirmed by genotyping. For liquid culture with Stomidazolone (AYSJ929, compound **3**), the analogs (AYSJ1059 (compound **5**) and AYSJ1061 (compound **6**)), degraded compound lzx103 (compound **4**), and SIM3* (compound **7**), all compounds were dissolved in dimethyl sulfoxide (DMSO, Molecular biology grade, Fujifilm Wako, Japan 046-21981), Col-0 seeds were sown in 96-well plates (TL5003; True Line) containing 95 µL of 1/2 MS media containing 0.5% sucrose with rotary shaking at 140 rpm under continuous light at 22 °C. Five µL of compounds were dissolved in DMSO at 10 mM and diluted with liquid 1/2 MS media to 1 mM and added on 1-day-old seedlings (final concentration, 50 µM). Seedlings were incubated for 8 days, and at day 9 seedling abaxial cotyledons were observed through confocal microscopy. For liquid culture of *scrm-D_S343* and *scrm-D_S423*, 1/2 MS liquid media without sucrose was used. For the solid culture, 0, 20, 50 or 100 µM Stomidazolone were solidified with low-melting agarose (5805S, PrimeGelTM Agarose LMT 1-20 K, TAKARA Bio). Briefly, 1/2 MS salt, 0.5% sucrose, and 0.8% low melting agarose were mixed in water and autoclaved. After cooling the medium at around 30 °C, Stomidazolone (500x, dissolved in DMSO) was added and poured into 90 mm petri dishes (Asnol Petri Dish JP φ90 x 15 mm, 3-1491-01, AS ONE). Col-0 seeds were sown on Stomidazolone-containing 1/2 MS agarose media and grown under continuous light for 9 days.

### Molecular cloning and generation of transgenic plants

The following constructs for transgenic plants were generated in this study: *MUTEpro::MUTE-GFP* (pAH11), *MUTEpro::MUTE$_{E162AE163A}$-GFP* (pHS133), *MUTEpro::MUTE$_{R133AR134A}$-GFP* (pHS134). The original *MUTEpro::MUTE-GFP* construct (pLJP155) was generated via a different cloning strategy with different antibiotics selection[11]. We therefore re-generated the wild-type Three-way Gateway version of *MUTEpro::MUTE-GFP* (pAH11) as a control. *MUTE* genomic coding construct without stop codon (pAR205) and *MUTE* promoter construct (pAR202) were subjected to the Gateway reaction with R4pGWB504[56]. The wild-type MUTE-GFP lines with the identical R4pGWB504 backbone enable a proper comparison of phenotypic complementation. The mutant versions of *MUTE* constructs were generated in the same way after site-

directed mutagenesis using Q5 Site-directed mutagenesis kit (NEB) and subjected to the Three-Way Gateway reaction after sequence confirmation. The constructs were then transformed into the *Agrobacterium* strain GV3101 and floral dipped into *mute-2/+* plants. To construct *CaMV35S::H2B-mScarlet-I3/pPZP211* (DKv1400), 35S promoter (*35Sp::RFP-CenH3* with *DKp665-DKp1254*)[57], H2B-(SGGGG)2 linker (from DKv1200[58] with DKp2412-DKp2599), mScarlet-I3 (from codon optimized synthetic DNA for *A. thaliana*), Nos terminator (from DKv1200 with DKp1251-DKp2598) were amplified using KOD one polymerase (TOYOBO). The PCR products were assembled into BamHI/EcoRI-digested pPZP211[59] using the NEBuilder HiFi DNA Assembly (NEB). For detailed lists of oligo DNA and plasmid information, see Tables S5 and S6, respectively.

### Confocal microscopy

For confocal microscopy, cell peripheries of seedlings were visualized with propidium iodide (Sigma, P4170). Images were acquired using LSM800 or LSM700 (Zeiss) using a 20x lens. The GFP, YFP reporter and PI signals were detected with excitation at 488 nm and an emission filter of 410 to 546 nm, at 488 nm and an emission filter of 410 to 546 nm, and with excitation at 561 nm and 582–617 nm emission range, respectively. Raw data were collected with 1024 × 1024 pixel images. For qualitative image presentation, Adobe Photoshop CC was used to trim and uniformly adjust the contrast/brightness.

### Yeast two hybrid (Y2H) assay

Y2H assays were performed using the Matchmaker™ Two-Hybrid System (Clontech). Bait (pGBKT7) and prey (pGADT7) constructs were co-transformed into the yeast strain AH109 using Frozen-EZ Yeast Transformation II Kit (Zymo Research, T2001). The resulting transformants were spotted on SD ( − Leu, −Trp) and SD ( − Trp, −Leu, −His) selection media containing different concentrations of 3-amino-1,2,4-triazole (Sigma, A8056) as previously reported[21]. Quantitative α-galactosidase assay was also performed according to manufactural instruction (Clontech). Briefly, the transformants were incubated in 2 ml SD medium with 0, 10, 50 100 µM Stomidazolone (**3**) or SIM3* (**7**) with rotary shake (220 rpm) at 30 °C overnight. The cultured media containing yeast were transferred to microtubes and centrifuged to obtain the supernatant. The colorimetric assay was performed by incubating 8 µL of supernatant with 24 µL Assay Buffer that contains 0.5 M sodium acetate (pH 4.5) (Nakalai tesque, 31137-25): 100 mM *p*-nitrophenyl α-D-Galactopyranoside Solution (Sigma N0877) = 2:1 (v/v) at 30 °C for 60 min, then terminated by adding 960 µL 1X Stop Solution (0.1 M sodium carbonate, Fujifilm Wako 195-01582). After OD$_{410}$ was recorded, α-galactosidase units were calculated according to the protocol (Clontech, P3024-1).

### Biomolecular fluorescence complementation (BiFC)

For ratiometric BiFC assays[60], the cDNAs of SPCH, MUTE, and FAMA were cloned into the 35S CaMV promoter-driven destination vector pSPYNE, which contains the N terminus of the EYFP protein (nYFP-174 amino acids), while the cDNA of SCRM and SCRM2 was inserted into pSPYCE, which contains the C terminus of the EYFP protein (cYFP-64 amino acids) (see Supplementary Table 5 for detailed information of the constructs). The constructs were then transformed into *Agrobacterium tumefaciens* strain GV3101. The Agrobacterium was resuspended in infiltration buffer (10 mM MgCl$_2$, 10 mM 2-(N-morpholino) ethanesulfonic acid [pH 5.6], and 150 µM acetosyringone). Bacterial culture densities were adjusted to a final optical density (OD$_{600}$) of 1.0, and the cell suspensions were incubated at room temperature for 4 hours prior to infiltration. Equal volumes of cultures carrying the corresponding complementary pair of BiFC constructs (YFPn and YFPc), along with *CaMV35S::Histone H2B-mScarlet-I3* and the silencing suppressor plasmid p19 (a gift from Dr. Sir David Baulcombe) were mixed and then co-infiltrated into 4-week-old *N. benthamiana* leaves.

The infiltrated *N. benthamiana* leaves were incubated at 25 °C for 2 days. Subsequently, leaf discs were cut out and cultured in mock or in liquid 1/2 MS media in mock (DMSO) or 100 μM Stomidazolone for 1 day. Confocal imaging of *N. benthamiana* leaves was performed using a Leica SP8 confocal microscope using a 20X objective lens (Leica Microsystems), simultaneously capturing YFP (excitation at 518 nm and emission at 530 nm) and mScarlet-I3 (excitation at 569 nm and emission at 600 nm) differential interference contrast channels.

## Chemical synthesis

Synthesis of Stomidazolone (AYSJ929: **3**): 4 Å Molecular sieve powder (500 mg) was added into a large screw-capped tube and heat-dried in vacuo for 1 h. 2,4-Diphenyloxazole (111 mg, 0.50 mmol), Chloramine B dihydrate (250 mg, 1.0 mmol, 2.0 equiv) and CuI (9.5 mg, 50 μmol, 10 mol%) were added into the tube with a magnetic stirring bar in open air. The tube was evacuated and refilled with $N_2$ gas following the usual Schlenk technique. Anhydrous 1,2-dichloroethane (5.0 mL, 0.10 M) was added to the tube via the rubber top, the tube capped and its cap wrapped with parafilm. The reaction mixture was stirred and heated at 40 °C for 12 h. The mixture was then cooled to room temperature, and the crude mixture was filtered through a pad of silica gel topped with $Na_2SO_4$ in a short column and concentrated *in vacuo*. Purification by flash chromatography on silica gel (hexane/ethyl acetate = 4:1) followed by recrystallization from $CH_2Cl_2$/methanol provided Stomidazolone (**3**) in 42% yield (113 mg, 0.21 mmol) as a colorless crystal. For details of chemical synthesis see Supplementary Data 1. For synthesis of biologically inactive Stomidazolone analogs AYSJ1059 (**5**) and AYSJ1061 (**6**) as well as the major inactive degradation product lzx103 (**4**) under alkaline condition, see Supplementary Data 1.

## NMR and mass-spectrometry

The high-resolution mass spectra were recorded on Thermo Fisher Scientific Exactive Plus (ESI) with Thermo Scientific Dionex UltiMate 3000 Series UHPLC or Agilent Technologies 6130 Quadrupole Liquid Chromatography/Mass Spectrometry (LC-MS). Nuclear magnetic resonance (NMR) spectra were recorded on JEOL JNM-ECA-600 (¹H 600 MHz, ¹³C 150 MHz) spectrometer. Chemical shifts for ¹H NMR are expressed in parts per million (ppm) relative to residual $CHCl_3$ in $CDCl_3$ (δ 7.26 ppm), and for ¹³C NMR in ppm relative to $CDCl_3$ (δ 77.2 ppm). Data are reported as follows: chemical shift, multiplicity (s = singlet, d = doublet, dd = doublet of doublets, t = triplets, dt = doublet of triplets, td = triplet of doublets, tt = triplet of triplets, q = quartets, m = multiplets, brs = broad singlet), coupling constant (Hz), and integration. For the original NMR and mass-spectrometry data, see Dataset S1.

## X-ray crystallography

A suitable crystal was mounted with mineral oil on a MiTeGen Micro-Meshes and transferred to the goniometer of the kappa goniometer of a RIGAKU XtaLAB Synergy-S system with 1.2 kW MicroMax-007HF microfocus rotating anode (Graphite-monochromated Mo Kα radiation (λ = 0.71073 Å)) and HyPix6000HE hybrid photon-counting detector. Cell parameters were determined and refined, and raw frame data were integrated using CrysAlis^Pro (Agilent Technologies, 2010). The structures were solved by direct methods with SHELXT[61]

and refined by full-matrix least-squares techniques against $F^2$ (SHELXL-2018/3) by using Olex2 software package[62]. The intensities were corrected for Lorentz and polarization effects. The non-hydrogen atoms were refined anisotropically. Hydrogen atoms were placed using AFIX instructions. CCDC 2089650 contains the supplementary crystallographic data for this paper. These data can be obtained free of charge from The Cambridge Crystallographic Data Centre via www.ccdc.cam.ac.uk/data_request/cif. See Supplementary Table 1 for X-ray crystallography data and the CIF Report.

## Liquid chromatography-mass spectroscopy (LC/MS)

Stomidazolone in 1/2 MS liquid media (pH 5.7) or phosphate buffer (pH 7.0) was detected by LC (Agilent 1200, Agilent Technologies, Santa Clara, USA) coupled to a single quadrupole mass spectrometer (Agilent 6130) equipped with an APCI-ES source. For the stability assay of Stomidazolone, the analytical samples were prepared as follows: 100 μL 5 mM Stomidazolone in DMSO was diluted with 900 μL 1/2MS (pH5.7) or 50 mM Phosphate buffer (pH 7.0). The Stomidazolone solutions were incubated at room temperature with rotating. At 0 or 48 hr after incubation, the 10 μL of solution were collected, diluted with water: acetonitrile;= 50:50, and applied to LC/MS. Stomidazolone (**3**) and the degraded compound lzx103 (**4**) were separated by Agilent Poroshell 120 EC-C18 (2.1 × 100 mm, 2.7) column. The mobile phase consisted of 5 mM ammonium formate: acetonitrile = 80:20 at a flow rate of 0.6 mL/min. in isocratic mode, and detected by mass spectrometry (MS). For the detection of these compounds, 0.5 μL of the sample were injected, and the total run time was 7 min. Parameters were set as follows: drying gas flow 13.0 L/min, nebulizer pressure 3100 Torr, drying gas temperature 350 °C, and capillary voltage +4000 V (positive ionization) and 3500 V (negative ionization). The electrospray ionization mass spectra (ESI-MS) recorded in the mass-to-charge ratio range from m/z 80–1000 was collected. For identification of Stomidazolone and lzx103, selected ion monitoring (SIM) mode was used, and the following m/z values were used to detect the different protonated molecules $[M + H]^+$ at m/z 532 for Stomidazolone, and m/z 261 for lzx103. Identification of Stomidazolone and lzx103 was exclusively based on LC retention time and high-resolution mass spectra.

## Recombinant protein expression and purification

*A. thaliana* SCRM (1-494), ᴀₙSPCH (98-364), MUTE (1-202), ᴀₙFAMA (190-414), MUTE_ΔC (1-114), MUTE_R133A_R134A, MUTE_E162A_E163A, MUTE_R133A_R134A_E162A_E163A, SCRM-ACTL (405-494), MUTE-ACTL (114-202), SPCH-ACTL (285-364), FAMA-ACTL (306-414), MUTE_ACTL_R133A_R134A, MUTE_ACTL_E162A_E163A, MUTE_ACTL_R133A_R134A_E162A_E163A were cloned into pGEX-4T-1 vector with an N-terminal GST tag and a thrombin cleavage sequence. For protein expression, the constructs were transformed into *E. coli* strain BL21. For each transformant, a single colony was selected and incubated in 10 ml LB liquid media. The overnight-incubated *E. coli* suspensions were transferred to 1 L LB medium and incubated at 37 °C for around 2 h until the $OD_{600}$ reached 0.4–0.6. The isopropyl β-D-1-thiogalactopyranoside (0.25 μM of final concentration) was added to the cultures and the strains were incubated at 22 °C for a further 18 h. GST-fused proteins were purified using glutathione agarose resin. The soluble portion of the cell lysate was loaded onto a GST-Sepharose column. Nonspecifically bound proteins were removed by washing the column with 20 mM Tris pH 8.0, 200 mM NaCl. The bound GST-fused protein was eluted with 10 mm glutathione and 20 mM Tris pH 8.0, 200 mM NaCl. The GST-fused proteins were exchanged with phosphate-buffered saline buffer, and then the solution was treated with 50 μg of thrombin for 10–12 h at 16 °C. The GST portion of the protein was cleaved during thrombin digestion, and then the whole solution was reloaded onto the glutathione *S*-transferase column to obtain pure protein. The purified proteins were further purified by gel filtration on a Superdex-200 (GE) column using fast protein liquid chromatography and phosphate buffer (pH 7.2) as the eluent. The purity

of the protein was checked by SDS-PAGE. For detailed lists of oligo DNA and plasmid information, see Supplementary Tables 5, 6, respectively.

## Isothermal titration calorimetry (ITC)
Binding of the small molecule Stomidazolone (3) to SCRM, MUTE, SPCH, FAMA, MUTE$_{\Delta C}$, MUTE_ACTL, SCRM_ACTL, SPCH_ACTL and FAMA_ACTL, MUTE$_{R133A\_R134A}$, MUTE$_{E162A\_E163A}$, MUTE$_{R133A\_R134A\_E162A\_E163A}$, as well as SCRM binding to MUTE in the presence and absence of Stomidazolone (3) / AYSJ1059 (5) /AYSJ1061 (6) was characterized at 30 °C using a Malvern PEAQ-ITC microcalorimeter. All the protein samples were dialyzed overnight using PBS buffer containing 2% DMSO and 1 mM DTT. Stomidazolone was dissolved in PBS buffer containing 2% DMSO and 1 mM DTT. Titrations were performed by injecting $1 \times 0.5$-µL and $12 \times 3$-µL or $1 \times 0.75$-µL and $18 \times 2$-µL or $1 \times 0.2$-µL and $25 \times 1.3$-µL aliquots of 1 mM Stomidazolone / AYSJ1059 / AYSJ1061 to 5-30 µM of protein in PBS buffer pH 7.4, containing 2% DMSO and 1 mM DTT. All titrations were carried out at least twice. The raw data were corrected using buffer and protein controls and analyzed using the software supplied by the manufacturer. See Supplementary Table 3 for details.

## Biolayer interferometry (BLI)
The binding affinities of SCRM to MUTE / MUTE$_{R133A\_R134A}$ / MUTE$_{E162A\_E163A}$ / MUTE$_{R133A\_R134A\_E162A\_E163A}$ in presence and absence of Stomidazolone (3) and its analogues (AYSJ1059 (5) and AYSJ1061 (6)), and Stomidazolone / AYSJ1059 / AYSJ1061 interactions with GST-tagged SCRM, MUTE, SPCH, FAMA, MUTE ΔACTL, MUTE_ACTL, SCRM_ACTL, SPCH_ACTL and FAMA_ACTL, MUTE$_{R133A\_134A}$, MUTE$_{E162A\_E163A}$, MUTE$_{R133A\_R134A\_E162A\_E163A}$ were measured using the Octet Red96 system (ForteBio, Pall Life Sciences) following the manufacturer's protocols. The optical probes coated with anti-GST were first loaded with 500 nM GST tagged proteins before kinetic binding analyses. The experiment was performed in 96-well plates maintained at 30 °C. Each well was loaded with 200 µL reaction volume for the experiment. The binding buffer used in these experiments contained 1× PBS supplemented with 2% DMSO, 0.02% Tween20. The concentrations of the Stomidazolone as the analyte in the binding buffer were 100, 50, 25, 12.5, 6.25, 3.12, and 1.56 µM. There was no binding of the analytes to the unloaded probes as shown by the control wells. Binding kinetics to all seven concentrations of the analytes were measured simultaneously using default parameters on the instrument. The data were analyzed using the Octet data analysis software. The association and dissociation curves were fit with the 1:1 homogeneous ligand model. The $k_{obs}$ (observed rate constant) values were used to calculate $K_d$, with steady-state analysis of the direct binding. See Supplementary Table 3 for details.

## Protein structural modelling
MUTE and MUTE ACTL domain homology models were used from Alpha fold program (https://alphafold.ebi.ac.uk/entry/Q9M8K6) The MUTE ACTL domain was subjected to constrained energy minimization to allow the global energy minimization and structural analysis using the AMBER 12 suite and VADAR[63,64]. Molecular docking of both racemic forms of Stomidazolone (3), AYSJ1059 (5), and AYSJ1061 (6) to the MUTE ACTL domain monomers was carried out using the high ambiguity-driven biomolecular docking (HADDOCK) approach[65,66]. Ambiguous interaction restraints (AIRs) were selected based on mutagenesis data and preliminary AUTODOCK model[67]. The pairwise ligand interface RMSD matrix over all structures was calculated, and the final structures were clustered using an RMSD cut-off value of 3.5 Å for both ligand and protein. The clusters were then prioritized using RMSD and the HADDOCK score (weighted sum of a combination of energy terms). See Supplementary Data 2 for PDB Coordinates and validation files of MUTE ACTL domain- Stomidazolone binding for both racemic enantiomers. See Supplementary Table 4 for binding energies for Stomidazolone-MUTE docking simulation.

## Measurement of stomatal opening in *Commelina benghalensis*
The experiments were performed as described previously[30,68]. Briefly, Stomidazolone or ABA were added to a multi-well plate with basal buffer [5 mM MES/bistrispropane (pH 6.5), 50 mM KCl, and 0.1 mM CaCl$_2$] to be diluted to a final concentration of 50 or 20 µM, respectively. *C. benghalensis* plants were incubated in the dark overnight; then, 4 mm diameter leaf discs were excised from fully expanded leaves using a hole punch (Biopsy Punch, Kai Medical). Samples were immersed in a basal buffer containing compounds for 30 min before being exposed to light (150 µmol m$^{-2}$ s$^{-1}$ red light and 50 µmol m$^{-2}$ s$^{-1}$ blue light) for 3 h. The stomatal apertures were observed using a stereoscopic microscope (BX43; Olympus). The microscopic images were acquired with a charge-coupled device (CCD) camera (DP27; Olympus), followed by autofocus image processing using cellSens standard software (Olympus). Measurement of stomatal apertures of *C. benghalensis* was performed by using an automation program, *DeepStomata*[69].

## Quantification and statistical analysis
For quantitative analysis of the abaxial leaf epidermis, confocal images were taken on the days after germination as indicated in the Figure legends. Preparation of images was done as described previously[32,70]. For counting epidermal cell types, stomatal density, stomatal precursor cells (meristemoids), total stomata-lineage cells (stomata and meristemoids) were calculated by counting cell types in an area of 0.4 mm$^2$ (0.63 mm × 0.63 mm) and plotted as per mm$^2$. A series of Z-stack confocal images was obtained to obtain images covering the epidermis and capturing GFP/YFP signal from the reporter lines. The area and the number of epidermal cells were quantified by using FIJI-ImageJ. Statistical analyses were performed using R ver. 4.0.2. Sample sizes (n) are indicated and defined in each legend. For the multiple sample comparison, 1-way ANOVA was performed followed by a post hoc Tukey HSD test. For the two-sample comparison, Student's T-test was performed. Graphs were generated using R ggplot2 package. For all boxplots, each box represents IQR (interquartile range), in which top, middle line, and bottom indicating 75, 50, and 25 percentiles, respectively, and the bar representing maximum to minimal values. Each magenta dot represents jittered individual data point. The datapoints that go beyond the bar are outliers, which are indicated as black dots. If the data point is greater than Q3 + 1.5*IQR (above the third quartile is greater than 1.5 times of the interquartile range) or less than Q1 − 1.5*IQR, that data point is classified as outlier. The color of Boxplot graph was based on ColorBrewer.org. The value of n, the number of each experiment or samples, the means of error bars, and how statistical significance was defined are indicated in a relevant figure legend. See Source Data for quantitative data underlying each figure.

## Reporting summary
Further information on research design is available in the Nature Portfolio Reporting Summary linked to this article.

# Data availability
Crystallographic data for the structure reported in this article have been deposited at the Cambridge Crystallographic Data Centre, under deposition numbers CCDC 2089650. Primer oligo DNA information and plasmid information are provided in Tables S5 and S6, respectively. Supplementary Data and Source Data are provided with this paper. Any additional data will be provided by the corresponding authors upon request. Materials generated and used in this study will be available from Keiko Torii (ktorii@utexas.edu). The synthesized

imidazolones (Stomidazolone; ID AYSJ929, compound **3**) and biologically inactive analogs (AYSJ1059 (**5**) and AYSJ1061 (**6**)) may be limited in quantity. Inquiries for availability could be directed to Kei Murakami (kei.murakami@kwansei.ac.jp). Source data are provided with this paper.

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

## Acknowledgements

We thank Amanda Rychel and Alex Hofstetter for constructing the 3-Way Gateway version of *MUTEpro::MUTE-GFP*, Nagoya ITbM Molecular Structure Center for assistance in NMR and mass-spectrometry, Miki Muranaka for plant care, Jason McLellan for the access to ITC, and Kosuke Watanabe for assistance with X-ray crystal structure analysis. We appreciate Satoshi Matsubara and Tokihiro Furuhashi for their advice on the identification of lzx103. We especially thank Yuichiro Tsuchiya for the insightful discussion and generously offering the lab space/facility for A.N., and Eundeok Kim for comments. A part of this work was conducted in Institute for Molecular Science supported by ARIM (JPMXP1223MS5012). This work was supported by MEXT/JSPS KAKENHI (JP16H01237, JP17H06476 and JP19H00990) to K.U.T.; Japan Science and Technology Agency (JST) PRESTO (JPMJPR20D8), TOBE MAKI Scholarship Foundation, and Tatematsu Foundation to K.M.; and ITbM Research Award to S.J.Y., A.N., and S.-K.H. A.N. was in part supported by the JST CREST Award (JPMJCR1924). S.J.Y. was supported by the JSPS Research Fellowship for Young Scientists DC1. S.-K.H. and D. K. were supported by the Young Leader Cultivation Program from the Institute of Advanced Research, Nagoya University. K.U.T. acknowledges the support from the Howard Hughes Medical Institute and Johnson & Johnson Centennial Chair of Plant Cell Biology at UT Austin.

## Author contributions

A.N., K.M., S.J.Y., S.-K.H and K.U.T. conceived the study. K.U.T., K.M., S.J.Y., S.K-H., A.N., and K.M.S. designed the experiments. A.N., K.M.S., S.J.Y., H.S., C.M.C., K.H., Z.L., H.K., R.I, S.K., Y.A., K.U.T. performed the experiments (forward chemical genetics/transgenic/phenotypic analysis A.N., S.J.Y., H.S., S.-K.H., C.M.C., H.K., K.U.T.; yeast/biochemical/biophysical experiments, K.M.S., R.I.; BiFC, H.S.; chemical synthesis, S.J.Y., K.H., Z.L., K.M.; structural analyses/modeling, K.M., K.M.S., Y.S.; stomatal movement assays, Y.A., T.K.). D.K. developed a codon-optimized plasmid; K.I. supervised S.J.Y.; A.N., K.M.S., H.S., K.M., Y.A. and K.U.T. performed data analysis and visualization. K.U.T. wrote the manuscript draft. All authors contributed to the editing of the manuscript. K.M. and K.U.T. are responsible for project administration and funding acquisition.

## Competing interests

The authors declare no competing interests.
