## [Peer Review File · Nature Communications]

REVIEWER COMMENTS

Reviewer #1 (Remarks to the Author):

Comments for the authors:

In this manuscript, the authors have identified a novel chemical compound, Stomidazolone, which acts as an inhibitor of stomatal differentiation. Stomidazolone achieves this by specifically binding to the C-terminal ACT-Like (ACTL) domain of MUTE, a pivotal basic-helix-loop-helix (bHLH) protein that is crucial for stomatal development. This interaction impedes the heterodimerization of MUTE with its bHLH partner SCREAM, culminating in the cessation of stomatal stem cell differentiation. Moreover, the authors have ingeniously engineered Stomidazolone-resistant MUTE variants and have corroborated their functionality *in vivo*. While the research presents significant findings, there are several major concerns that need to be addressed to strengthen the conclusions drawn from the study:

1. Stomidazolone exhibits binding not only to MUTE but also to SPCH, FAMA, and SCRM proteins. Notably, in the *scrm* mutant, treatment with Stomidazolone resulted in the phenotype similar to that of *spch*, indicating that Stomidazolone inhibits SPCH activity in the absence of SCRM. Therefore, the conclusion that "Stomidazolone disrupts the heterodimerization of MUTE with SCRM to inhibit stomatal development" is inaccurate.
2. The function of *scrm-DL484F* and *scrm-DDACTL* should be introduced in this paper. Previous research conducted by the same group. (2022, PNAS) has shown that these two mutation versions of SCRM prevent heterodimerization with MUTE. If Stomidazolone inhibits stomatal development by disrupting the interaction between SCRM and MUTE, then the mutants *scrm-D_S343* and *scrm-D_S423* should be unaffected by Stomidazolone treatment. However, in the figure 4, these two mutants still response to Stomidazolone treatment. Additionally, it should compare the stomatal phenotypes of wild type and these two mutants with or without Stomidazolone treatment.
3. In the Y2H assay presented in Figure 5, Stomidazolone was found to inhibit the heterodimerization of SCRM with SPCH, MUTE, and FAMA, with the most significant inhibitory effects observed on the interaction between SCRM and FAMA. These results appear to contradict the high binding affinity of Stomidazolone for MUTE compared to the other proteins tested. Y2H only is not enough to support the conclusion that Stomidazolone inhibits the heterodimerization of SCRM with various partners, the authors need to provide more evidence, such as the CoIP in plants. Additionally, it will be interesting to test the effects of Stomidazolone when the ACTL domain of MUTE were replaced with that of SPCH or FAMA.
4. The authors generated a serious of mutant viersions of MUTE, which have low binding affinity to Stomidazolone . What is the interaction o SCRM with these mutant versions of MUTE, and whether these the interaction are regulated by Stomidazolone?

5. In the discussion, authors state that their discovery brings several implications. One of them is that Stomidazolone could be optimized in the future to map the functional significance of ACTL domains among the bHLH TFs. But I have a query that whether this phenomenon that Stomidazolone binding bHLH TFs through ACTL domains is common. As there are great differences of Stomidazolone binding between MUTE, SPCH, FAMA, and SCRMs proteins, albeit they are sister bHLHs. So, the biological significance of Stomidazolone remains to be explored.
6. As HDG2 positively regulates stomatal differentiation by promoting MUTE expression, *hdg2 atml1* double mutant should be insensitive to Stomidazolone, which is similar with *mute* mutant. But *hdg2 atml1* double mutant exhibited hypersensitivity to Stomidazolone in fig. 3n and 3o. Besides, in fig. 3o, why does Stomidazolone decrease the stomata numbers and meristemoid numbers of *hdg2 atml1* epidermis while the total numbers of stomata and meristemoids were unaffected compared with Mock?
7. In fig.S3, the results showed that Stomidazolone could binding MUTE_ACTLE162A_E163A, but in fig.6e and 6f, MUTEpro::MUTEE162A_E163A-GFP exhibit Stomidazolone resistance, which is contradictory. The results that the number of meristemoid of MUTEpro::MUTE-GFP and MUTEpro::MUTEE162A_E163A-GFP response to Stomidazolone in fig. 6f is inconsistent with fig.S5c.
8. liquid MS cultures: All the experiments and their quantifications stem from liquid MS culture grown plants. In these experiments the seedlings are probably experiencing an osmotic shock, an osmotic control should be added to show that this is indeed Stomidazolone specific. While I understand that these are much easier to deal with, I am a bit worried about how submergence of plants affects stomatal differentiation and the sensitivity to Stomidazolone is obviously highly limited. Therefore, I would like to see how solid vs. liquid culture affects stomatal differentiation with Stomidazolone for wild type. In my opinion, expanding the chemical space for manipulating protein functions in plants, liquid culture is not a good choice.

Minor comments:

1. There should be statistical analysis in fig. 4f and 4h.
2. There should be wild type control in fig.3 and fig.4.
3. In fig.6f, the number of meristemoid is so few that the significance difference is not apparent. So, the Y-axis should be adjusted.
4. In Fig. 1d the numbers of stomata seem not to be affected in the existence of exogenous Stomidazolone, the case then this observation should be discussed.
5. There is a large variation in meristemoid density for the WT between (Fig. 1e) and (Fig. 2c) in the presence of 50 μ M Stomidazolone. Why?

Reviewer #2 (Remarks to the Author):

The authors report the identification and functional characterization of a new chemical tool for plant biology research. Subsequent mode-of-action studies reveal that the compound targets and thus impairs the function of a dimeric transcription factor critical for stomatal development. This is an unusual mode-of-action with significance even beyond plant biology. Overall, the author report interesting and exciting findings that in my view in principal warrant publication in a prestigious journal such as Nat Commun.

The MS is rich in data and contains methods and approaches from diverse areas of research. I am no expert in the field of stomatal plant development and I will therefore not comment on this part of the manuscript (e.g. I cannot judge if the described phenotypes match the expectations or if the mutant selection + experimental outcome are appropriate). I will focus on the (bio)chemical part of the MS, which is an important part of the study as it serves to adequately prove the mode-of-action of the compound and thus the general claim of the MS. Although I overall like the work, there are some issues that in my view need to be addressed prior to publication:

1) The authors report Stomidazolone as a new chemical tool. I could however imagine that Stomidazolone may be unstable and potentially degrade during the long-term assay conditions, e.g. via elimination of a sulfonamide moiety which would impact all down-stream analyses (in particular the biochemical measurements). Have the authors assured the chemical stability of the compound under assay conditions (e.g. in different buffers over several days?). These data should be added to the MS. The authors should also provide a better ¹³C NMR spectrum of the compound in the SI, the current copy features rather weak signal intensities and some signals can only be suspected. Finally, it should be mentioned in the MS right from the beginning that the compound is a racemic mixture (see also point 2 for this).

2) A drawback of the study is a lack of suitable controls that unambiguously show that the observed effects are caused by Stomidazolone-mediated modulation of MUTE function. The authors use SIM1 during the plant assays which is however not an optimal control as it is not inactive but displays an alternative activity in stomatal development (reported in a previous paper by the authors). They also do not employ any “inactive” control in the biochemical assays. In my view it will be essential to show that phenotypically inactive compounds are also inactive in the biochemical assays, while active compounds are in contrast active in both assay types.

An almost optimal control for these phenotypic and biochemical assays in my view are the two enantiomers of Stomidazolone. As the authors claim that a discrete molecular interaction is the molecular basis of the phenotype and as Stomidazolone is a very small molecule with a rather small and discrete 3D surface, both enantiomers should display different activities in the phenotypic screen as well as in the biochemical assays. They are “optimal controls” as both compounds, due to their stereoisomeric nature, display similar “unspecific effects”, e.g. plant distribution properties, etc.. Indeed, different stereoisomers are frequently used as controls in biological and biochemical assays (in which they are sometimes called eutomer and distomer).

The authors report that Stomidazolone is accessible by a simple one-step synthesis in more than 100 mg yield. It should therefore be feasible to perform a subsequent enantiomeric separation of the currently racemic mixture to obtain some mg material of the separated enantiomers and to then test them in parallel in the most important plant and biochemical assays (there is no need to redo all plant assays but they should show that one enantiomer induces the desired phenotypic effect while the other is inactive). I would also recommend to have a relook at the binding pose analysis underlying Fig. 6b as two enantiomers should not display equal binding poses.

3) I am also a little bit worried about the raw data of the ITC measurements. I appreciate that the authors have done these measurements as they confirm the molecular model underlying the mode-of-action of the compounds. The ITC measurements are also much more direct and meaningful than BLI measurements which may be influenced by many unspecific events and I acknowledge that the authors have done a lot of work here. However, an ITC Wiseman plot should be sigmoidal (and not almost linear as the plots depicted in Supp Fig. S3). Also important, an upper plateau level should be reached that indicates saturation of the binding site and thus specific binding. Currently, these plots do not look very convincing and seem to indicate unspecific effects, e.g. protein aggregation, etc.? Are the expressed proteins adequately folded? If yes, what model can explain the shape of these curves be explained (and allow data extraction from them)?

Besides these main points, some further minor points:

4) Although the MS is overall well written and logically structured, there seem to be some expressions/wordings that are perhaps better rephrased. Here only some examples – line 47 “vales” should read as “valves”?; line 140: “we took a chemical-genetic approach” – not sure, if this is a “classical” chemical genetic approach which is normally used to describe a screening approach with chemical libraries; the approach here is perhaps better described as “mechanistic studies with mutants”; line 114: “acute angular shape”?; line 79: “Stomata are not the exception” should read as “Stomata are no exception”?; “To expand a chemical space to control stomatal function...” – this sounds somehow strange. There are more examples, please have a critical relook at the text.

5) As stated above, I am no expert on the plant assays and I assume that they have been done with the necessary care and accurateness. I am only wondering on the statistical analysis (which again I assume is overall correct). I understand that the statistically difference group assignments (e.g. a, b, c in Fig. 3) are the outcome of the applied statistical analysis. However, should these assignments not also reflect somehow “obvious” statistical differences? For example, in Fig. 3d, all mock treated data points belong to group a and Stomidazolone-treated samples belong to group b (which makes sense). In Fig. 3f however, the mock treated samples belong to a, but also the t- and s-Stomadizolone samples belong to a, only the m-sample is b (which I find difficult to grasp by looking on the data distribution in the figure). In some measurements, samples even belong to “ab” groups? Again, I think that the authors know what they are doing here but it might be worth to add some explanatory words on this data analysis in the SI for interested readers (like me).

6) Please ensure that all figures are uniformly labeled, e.g. in Fig. 2c,d – y axis is labeled as “/mm²” while in Fig. 3, the y-axis is labeled as “/ 1 mm²”. I also find it difficult to distinguish the colors (green, cyan) in the box plots. Black dots indicate outliers? The outlier selection is based on which calculation?

We sincerely thank the two expert Peer Reviewers for insightful and thoughtful comments. We have fully revised our manuscript, incorporating their keen comments and suggestions.

For the revision, we exhaustively performed the following major experiments:

- i. Ratiometric BiFC experiments *in planta* showing that Stomidazolone selectively disrupts the heterodimerization of SCRM-MUTE (Revised Fig. 6c, d).
- ii. The effects of Stomidazolone supplemented in solid media on stomatal differentiation of seedlings (Revised Fig. S2).
- iii. The Stomidazolone insensitivity of *scrm-D_S343 scrm2-1* and *scrm-D_S423 scrm2-1* double mutants (Revised Fig. 5m-o).
- iv. Identification of biologically inactive Stomidazolone analogs: We synthesized and identified the biologically inactive Stomidazolone analogs and further demonstrated that these analogs do not bind to the MUTE ACTL domain by binding kinetics assays and docking modeling (Revised Figs. 2, S6-S8, Tables S2, S3, Document S1).
- v. Protein-protein interaction analysis of recombinant mutant versions of MUTE with SCRM, the effects of Stomidazolone on the heterodimers of SCRM-MUTE mutant versions, and the effects of inactive Stomidazolone analogs on SCRM-MUTE (Revised Fig. S8, Tables S2, S3).
- vi. Stability assays of Stomidazolone in solvent and in aqueous conditions showing that Stomidazolone is stable under our experimental conditions. We further identified the major degradation product and performed bioassays demonstrating that the major degradation product has no effect on stomatal differentiation (Revised Figs. S3, S4, Document S1).
- vii. Resynthesis of Stomidazolone and retake of NMR spectra (Revised Document S1).
- viii. Repurification of recombinant proteins and re-analyses by ITC (Revised Fig. S6, S7, Tables S2, S3).

We additionally improved data presentation and writing as requested by the Reviewers. All these new results strongly support and further strengthen our original discovery of Stomidazolone as an inhibitor of stomatal differentiation with unique mode of action on interfering bHLH heterodimers, primarily SCRM-MUTE. We now include researchers, Kota Hashimoto and Zixuan Li, who synthesized Stomidazolone analogs, and Daisuke Kurihara, who generated an optimized RFP protein construct for our BiFC experiments, as co-authors.

Please see our point-by-point responses below (in boldface). We have also provided our revised main text and supplementary text files with changes highlighted in yellow separately.

Reviewer #1 (Remarks to the Author):

Comments for the authors:

In this manuscript, the authors have identified a novel chemical compound, Stomidazolone,

which acts as an inhibitor of stomatal differentiation. Stomidazolone achieves this by specifically binding to the C-terminal ACT-Like (ACTL) domain of MUTE, a pivotal basic-helix-loop-helix (bHLH) protein that is crucial for stomatal development. This interaction impedes the heterodimerization of MUTE with its bHLH partner SCREAM, culminating in the cessation of stomatal stem cell differentiation. Moreover, the authors have ingeniously engineered Stomidazolone-resistant MUTE variants and have corroborated their functionality *in vivo*. While the research presents significant findings, there are several major concerns that need to be addressed to strengthen the conclusions drawn from the study:

We truly appreciate Reviewer 1 for recognizing that our work presents significant findings.

1. Stomidazolone exhibits binding not only to MUTE but also to SPCH, FAMA, and SCRM proteins. Notably, in the *scrm* mutant, treatment with Stomidazolone resulted in the phenotype similar to that of *spch*, indicating that Stomidazolone inhibits SPCH activity in the absence of SCRM. Therefore, the conclusion that "Stomidazolone disrupts the heterodimerization of MUTE with SCRM to inhibit stomatal development" is inaccurate.

As Reviewer 1 astutely points out, we observed that Stomidazolone application to the *scrm* mutant results in a *spch*-like phenotype. SCRM and its paralog SCRM2 are known to exhibit unequal redundancy when partnering with SPCH, MUTE, and FAMA (Kanaoka et al. 2008 Plant Cell). Whereas the *scrm2* mutant exhibits no discernable stomatal phenotype, it drastically exaggerates the *scrm* mutant phenotype (i.e., *scrm scrm2* double mutant resembles *spch*). Thus, it may be possible that the interaction of the SCRM2-SPCH heterodimer is weaker than that of SCRM-SPCH, and as such, in the absence of SCRM, the SCRM2-SPCH heterodimer can also be perturbed by Stomidazolone *in vivo*. This hypothesis explains the observed effects of Stomidazolone on the *scrm* mutant.

During the revision period, we made exhaustive efforts to test our hypothesis. However, these efforts were hampered by the unstable nature of the recombinant SCRM2 protein, making it technically impossible to produce sufficient amounts of SCRM2 protein to test our hypothesis.

To accurately reflect all our observations, we have adjusted the conclusion to state that "Stomidazolone primarily disrupts the heterodimerization of MUTE with SCRM to inhibit stomatal development" throughout our manuscript. Additionally, we now provide our interpretation regarding SCRM2-SPCH heterodimerization in the revised Results and Discussion sections (page 8, 3rd paragraph page 13, last paragraph). Thank you very much for the insightful critique.

2. The function of *scrm*-DL484F and *scrm*-DDACTL should be introduced in this paper. Previous research conducted by the same group. (2022, PNAS) has shown that these two mutation versions of SCRM prevent heterodimerization with MUTE. If Stomidazolone inhibits stomatal development by disrupting the interaction between SCRM and MUTE, then the mutants *scrm*-D_S343 and *scrm*-D_S423 should be unaffected by Stomidazolone treatment. However, in the figure 4, these two mutants still response to Stomidazolone treatment.

We concur with Reviewer 1 that our original descriptions of *scrm-D* intragenic suppressor mutants were brief, which may have led to confusion regarding the interpretation of their phenotypes and Stomidazolone resistance.

As Reviewer 1 points out, we previously reported these two alleles as intragenic suppressors of *scrm-D* (Seo et al. 2022 PNAS). In the 'wild-type' background, these suppressor alleles exhibit stomatal patterning and stomatal index nearly indistinguishable from that of wild type (see Fig.a below, from Seo et al. 2022).

Fig. a. Intragenic suppressor alleles of *scrm-D* exhibit nearly normal epidermis with stomatal index comparable to that of wild type. -Figures reproduced from Seo et al. 2022 PNAS

These alleles either remove or perturb the structural integrity of the ACTL domain (*scrm-D_S343* and *scrm-D_S423*, which code for *scrm-D_L484F* and *scrm-D Δ ACTL*, respectively). A striking property of these *scrm-D* ACTL mutants is that while they can form functional heterodimers with SPCH to initiate stomatal cell lineages, they are unable to form heterodimers with MUTE. However, this specific interaction property of *scrm-D* ACTL mutants can be masked by the presence of SCRM2, which can also heterodimerize with SPCH/MUTE/FAMA to direct stomatal differentiation. Therefore, *scrm-D_S343* and *scrm-D_S423* are 'sensitized' mutants in regard to the regulation of MUTE activity.

In our original manuscript, we tested our hypothesis that Stomidazolone impacts SCRM2-MUTE heterodimerization by treating Stomidazolone to *scrm-D_S343* and *scrm-D_S423* genotypes that possess the functional SCRM2 allele. If Stomidazolone disrupts SCRM2-MUTE heterodimers, it will increase the arrested meristemoids in *scrm-D_S343*

and *scrm-D_S423*, which otherwise develop nearly normal epidermis (see Fig. a). Our results, originally presented in Fig. 4e-h, fully support our prediction. Unlike the *scrm* mutant (see Comment 1 above), however, neither *scrm-D_S343* nor *scrm-D_S423* exhibits a *spch*-like phenotype. Thus, the interaction of SPCH with *scrm-D_L484F* and *scrm-D Δ ACTL* is not perturbed by Stomidazolone.

As we reported in the Seo et al. paper, in the *scrm2* knockout background, these *scrm-D* *ACTL* mutants exhibit massive clusters of arrested meristemoids (See Fig. b below).

Fig. b. *scrm-D* *ACTL* mutations trigger arrested meristemoids in the absence of *SCRM2*. -Figures reproduced from Seo et al. 2022 PNAS

We presume that Reviewer 1 assumed that the *scrm-D_S343* and *scrm-D_S423*, which we used in our study, carry the additional *scrm2-1* mutation. If so, Reviewer 1 is correct that both *scrm-D_S343 scrm2-1* and *scrm-D_S423 scrm2-1* would be resistant to Stomidazolone treatment.

To address this hypothesis, during the revision period, we performed Stomidazolone treatment on *scrm-D_S343 scrm2-1* and *scrm-D_S423 scrm2-1* seedlings. As Reviewer 1 predicted, these seedlings are insensitive to Stomidazolone treatment, exhibiting meristemoid clusters. These new data are provided as Revised Fig. 5m-o and further discussed regarding the implications (page 13-14). Due to small, wrinkly cotyledons with massive arrested small meristemoids, it was not possible to provide cell counting). Our new data further highlight MUTE (SCRM-MUTE heterodimer) as Stomidazolone's primary target. Thank you so much for providing insightful comments.

Additionally, it should compare the stomatal phenotypes of wild type and these two mutants with or without Stomidazolone treatment.

Thank you for the thoughtful suggestion. We always include wild-type controls in all of our bioassay experiments. The control wild-type data were provided in Fig. 1 and Fig. 2 of our original submission. In response to Reviewer 1, we now provide the control wild-type data in Revised Fig. 5a, g, and m.

3. In the Y2H assay presented in Figure 5, Stomidazolone was found to inhibit the heterodimerization of SCRM with SPCH, MUTE, and FAMA, with the most significant inhibitory effects observed on the interaction between SCRM and FAMA. These results appear to contradict the high binding affinity of Stomidazolone for MUTE compared to the other proteins tested. Y2H only is not enough to support the conclusion that Stomidazolone inhibits the heterodimerization of SCRM with various partners, the authors need to provide more evidence, such as the CoIP in plants. Additionally, it will be interesting to test the effects of Stomidazolone when the ACTL domain of MUTE were replaced with that of SPCH or FAMA.

Thank you for raising such an important point. Reviewer 1 is correct that our Y2H assay shows the strong effects of Stomidazolone on SCRM-FAMA interactions in yeast. However, linking this data to the biological significance (plant phenotypes) may not be as straightforward as it seems. Importantly, the inhibition of SCRM-FAMA heterodimerization by Stomidazolone will not be seen *in vivo* because FAMA expression requires MUTE (FAMA is a direct MUTE target - See: Han et al. 2018 DEV CELL). Therefore, that Stomidazolone interferes with SCRM-FAMA does not necessarily translate into that FAMA is the *in vivo* target of Stomidazolone. This notion is consistent with our observation that a number of *fama* tumors are significantly reduced by Stomidazolone application (Fig. 2b, d of our original submission, now Fig. 3b, d). Additionally, whereas Y2H is a highly sensitive system, it has a drawback of utilizing reporter gene expression as a readout of protein-protein interactions. For example, it is well known that transcription factors (TFs) often result in autoactivation in Y2H due to the transcriptional activation domain existing in the TF of interest. Because both SPCH and FAMA cause strong autoactivation in yeast, we used their N-terminal-deleted versions (*i.e.* removing their activation domains) for our Y2H assays to avoid this problem.

As keenly suggested by Reviewer 1, during the revision period, we performed *in planta* ratiometric Biomolecular Complementation assays (BiFC) using *Nicotiana benthamiana*. The system includes a nuclear RFP marker (HistoneH2B-mScarlet-I3) and pairwise combinations of split YFP constructs fused to the full-length stomatal bHLH proteins (SCRM-cYFP alone, or SCRM-cYFP with SPCH-nYFP, MUTE-nYFP, or FAMA-nYFP), all of which are co-transfected into developing *N. benthamiana* leaves. Subsequently, leaf disks were treated either in a mock (DMSO) or in the presence of Stomidazolone. YFP signals were normalized against RFP signals (see revised Methods section). Dr. Daisuke Kurihara, who constructed a unpublished codon-optimized HistoneH2B-mScarlet-I3 construct is now included as a co-author.

As shown in Revised Fig. 6c and d, Stomidazolone application severely perturbed the SCRM-MUTE YFP signals, indicating that Stomidazolone interferes with SCRM-MUTE heterodimerization *in planta*. By contrast, Stomidazolone exerted modest effects on SCRM-SPCH and no effects on SCRM-FAMA heterodimerizations. These results fully accord with our *in vivo* phenotypic characterizations (Figs. 1-3) and support the original conclusion that Stomidazolone primarily targets SCRM-MUTE *in vivo*. The differences in the efficacy of Stomidazolone for SPCH and MUTE may lie in their differential binding partners *in vivo*. In support of this idea, it is known that SPCH and MUTE associate with different transcriptional or epigenetic components (e.g. Matos et al. 2014 eLife; Kim et al. 2022 Nature Plants). We now discuss such possibilities in our revised manuscript (page 13, 2nd paragraph). Our oligo DNA and plasmid lists (Tables S5 and S6) have been updated as well.

We are convinced that our multi-faceted approaches of the biophysical *in vitro* binding kinetics assays, Y2H, BiFC *in planta*, exhaustively performed *in vivo* phenotypic analyses, as well as structure-guided engineering of Stomidazolone-resistant plants compellingly support the notion that Stomidazolone is a one of a kind inhibitor of stomatal differentiation with a unique mode of action to interfere with the heterodimerization of plant bHLH-ACTL TFs, preferentially SCRM-MUTE. We believe that further structural analyses, including domain-swaps, are beyond the scope of this manuscript.

4. The authors generated a series of mutant versions of MUTE, which have low binding affinity to Stomidazolone. What is the interaction of SCRM with these mutant versions of MUTE, and whether these interactions are regulated by Stomidazolone?

We sincerely appreciate Reviewer 1's excellent comments! In response to Reviewer 1, we analyzed the *in vitro* binding kinetics of SCRM heterodimerization with the three mutant versions of the MUTE protein. Consistent with our original report that mutant MUTE-GFP proteins driven by the *MUTE* promoter rescued *mute* phenotype (original Fig. 6, now Fig. 7), we found that these mutant MUTE proteins are still able to interact with SCRM. The binding affinity of these mutant MUTE with SCRM was reduced by about two to three-fold, suggesting that amino-acid substitutions within the MUTE_ACTL domain may affect heterodimerization strengths. Most importantly, Stomidazolone does not significantly affect the heterodimer formation of SCRM with the mutant versions of MUTE. These new results further support our major finding of Stomidazolone's mode of action in SCRM-MUTE heterodimerization. These data are now included in Fig. S8a-c, Table S3 of our revision.

5. In the discussion, authors state that their discovery brings several implications. One of them is that Stomidazolone could be optimized in the future to map the functional significance of ACTL domains among the bHLH TFs. But I have a query that whether this phenomenon that Stomidazolone binding bHLH TFs through ACTL domains is common. As there are great differences of Stomidazolone binding between MUTE, SPCH, FAMA, and SCRM proteins, albeit they are sister bHLHs. So, the biological significance of Stomidazolone remains to be explored.

We fully concur with Reviewer 1 that more structure-function analyses would be needed to conclusively state whether Stomidazolone binding to bHLH TFs through the ACTL domain can be expanded to other plant bHLH-ACTL proteins. In response to Reviewer 1's critique, we have modified our Discussion (p14, 2nd paragraph) to emphasize that further studies will clarify the exact biological significance of Stomidazolone.

6. As HDG2 positively regulates stomatal differentiation by promoting MUTE expression, *atml1* double mutant should be insensitive to Stomidazolone, which is similar with *mute* mutant. But *hdg2 atml1* double mutant exhibited hypersensitivity to Stomidazolone in fig. 3n and 3o. Besides, in fig. 3o, why does Stomidazolone decrease the stomata numbers and meristemoid numbers of *hdg2 atml1* epidermis while the total numbers of stomata and meristemoids were unaffected compared with Mock?

Thank you for the keen comments. Based on our original discovery (Peterson et al. 2013 Development), we think that *hdg2 atml1* double mutant is hypersensitive because the

available MUTE in this double mutant background is barely above the threshold (so *hdg2 atm1* is a 'sensitized' mutant with regard to interfering SCRM-MUTE heterodimerization). As for the original Fig. 3o box plot, after carefully going over the original quantitative data, we realized that we have inadvertently selected the incorrect Excel spreadsheet cell when generating a boxplot for *hdg2 atm1* mock meristemoid. We have corrected the errors (see revised Fig. 4o). We sincerely apologize for the oversight and wholeheartedly thank Reviewer 1 for catching it.

7. In fig.S3, the results showed that Stomidazolone could binding MUTE_ACTLE162A_E163A, but in fig.6e and 6f, MUTE_{pro}::MUTEE162A_E163A-GFP exhibit Stomidazolone resistance, which is contradictory. The results that the number of meristemoid of MUTE_{pro}::MUTE-GFP and MUTE_{pro}::MUTEE162A_E163A-GFP response to Stomidazolone in fig. 6f is inconsistent with fig.S5c.

Our ITC data, originally provided as Fig. S3, show that MUTE_{E162A_E163A} exhibits substantial reduction in Stomidazolone binding with a nearly 5-fold increase in the K_d value. We have repeated this ITC experiment in response to Reviewer 2's comments, and essentially obtained the same results (Revised Fig. S7, Tables S2, S3). Our finding that MUTE_{pro}::MUTE_{E162A_E163A} confers Stomidazolone resistance suggests that the observed reduction in *in vitro* binding affinity is sufficient for the *in vivo* developmental response. As for the comments on MUTE_{pro}::MUTE-GFP and MUTE_{pro}::MUTE_{E162A_E163A}-GFP in our original Fig. 6 and Fig. S5c, these are different transgenic lines. Whereas both lines for all constructs exhibit similar effects, there are always some variabilities in different transgenic lines. We have further repeated the experiments and now present a representative data in our revised figure (see Revised Fig. S11c). Thank you.

8. liquid MS cultures: All the experiments and their quantifications stem from liquid MS culture grown plants. In these experiments the seedlings are probably experiencing an osmotic shock, an osmotic control should be added to show that this is indeed Stomidazolone specific. While I understand that these are much easier to deal with, I am a bit worried about how submergence of plants affects stomatal differentiation and the sensitivity to Stomidazolone is obviously highly limited. Therefore, I would like to see how solid vs. liquid culture affects stomatal differentiation with Stomidazolone for wild type. In my opinion, expanding the chemical space for manipulating protein functions in plants, liquid culture is not a good choice.

We fully agree with Reviewer 1 that liquid culture may introduce potential complications in interpreting the stomatal phenotype. As suggested, during the revision period, we tested the effects of Stomidazolone supplemented in solid MS media in 4 different concentrations (0, 20, 50, and 100 μ M; see Methods). As shown in Fig. S2 of our revised manuscript, just like in liquid culture, Stomidazolone in the solid media inhibits stomatal differentiation in a dose-dependent manner, with concomitant increase in the meristemoid index. We also observed the overall seedling phenotypes grown on the Stomidazolone-containing MS solid media plates (Fig. S2d). Stomidazolone did not appear to affect seedling growth on 20 and 50 μ M, but caused subtle growth retardation at 100 μ M. These results suggest that Stomidazolone taken up from the roots can inhibit stomatal differentiation, thereby broadening the practical utility of Stomidazolone and facilitating future expansion of chemical space for manipulating bHLH-ACTL protein functions in plants. Thank you very much for the fantastic suggestion!

Minor comments:

1. There should be statistical analysis in fig. 4f and 4h.

As suggested, we are now providing the boxplot with statistical analysis. Please see Revised Fig. 5g-l.

2. There should be wild type control in fig.3 and fig.4.

Our original submission includes wild-type control data in Fig. 3a, b. As requested, we have now added yet another wild-type control as Fig. 4a, b, g, h, and m in our revised manuscript. Thank you.

3. In fig.6f, the number of meristemoid is so few that the significance difference is not apparent. So, the Y-axis should be adjusted.

Thank you for your suggestions. Given the volume of our data, we feel that providing yet additional graphs might overwhelm the readers. We added exact p values for comparing the meristemoid data with statistical significance. We are in the process of depositing the original spreadsheets of counting data to Texas Data Repository (<https://dataverse.tdl.org/> -the repository accession number will be provided) and we hope that would be sufficient. Thank you.

4. In Fig. 1d the numbers of stomata seem not to be affected in the existence of exogenous Stomidazolone, the case then this observation should be discussed.

We truly thank Reviewer 1 for catching this. For characterizing the meristemoid arrested phenotype, we observed 9-day-old after germination (dag) seedlings to ascertain that these meristemoids are most likely 'arrested'. For examining the expression of stomatal-lineage and stomatal-differentiation markers, we originally observed 7-dag seedlings for stronger, robust GFP/YFP signals (their signals diminish as cotyledons mature). It is most likely that in a few days, a fraction of the meristemoids expressing these markers will differentiate into stomata in mock. We realized that displaying the microscopy images of epidermis from different dag can be misleading. During the revision period, we re-did the experiments at 9 dag to be consistent with the phenotypic analysis in fig. 1c and 1e. The results support our original conclusion that Stomidazolone treatment reduces the number of stomata with concomitant increase of arrested meristemoids. Thank you so much for the keen suggestion.

5. There is a large variation in meristemoid density for the WT between (Fig. 1e) and (Fig. 2c) in the presence of 50 μ M Stomidazolone. Why?

These separate experiments were performed in different growth chambers in Nagoya, Japan due to the first author's job relocation. While we made every effort to replicate the growth conditions, this might have led to the observed variation. We would like to respectfully point out that the same growth chamber was used for the control wild-type and a series of stomatal mutants presented in Fig. 2 (Revised Fig. 3) to accurately delineate the specific action of Stomidazolone. Thank you.

Reviewer #2 (Remarks to the Author):

The authors report the identification and functional characterization of a new chemical tool for plant biology research. Subsequent mode-of-action studies reveal that the compound targets and thus impairs the function of a dimeric transcription factor critical for stomatal development. This is an unusual mode-of-action with significance even beyond plant biology. Overall, the author report interesting and exciting findings that in my view in principal warrant publication in a prestigious journal such as Nat Commun.

The MS is rich in data and contains methods and approaches from diverse areas of research. I am no expert in the field of stomatal plant development and I will therefore not comment on this part of the manuscript (e.g. I cannot judge if the described phenotypes match the expectations or if the mutant selection + experimental outcome are appropriate). I will focus on the (bio)chemical part of the MS, which is an important part of the study as it serves to adequately prove the mode-of-action of the compound and thus the general claim of the MS. Although I overall like the work, there are some issues that in my view need to be addressed prior to publication:

We sincerely thank Reviewer 2 for highly valuing our work, recognizing that our work may have broader significance beyond plant biology.

1) The authors report Stomidazolone as a new chemical tool. I could however imagine that Stomidazolone may be unstable and potentially degrade during the long-term assay conditions, e.g. via elimination of a sulfonamide moiety which would impact all down-stream analyses (in particular the biochemical measurements). Have the authors assured the chemical stability of the compound under assay conditions (e.g. in different buffers over several days?). These data should be added to the MS.

Thank you for your insightful comment. This is an important point. As requested, we have investigated the stability of Stomidazolone. Stomidazolone itself is a white solid and a stable compound. Using ¹H-NMR and high-resolution mass-spectrometry (HRMS), we have confirmed that this molecule is stable under acidic conditions but degraded under basic conditions. For technical reasons, these experiments were conducted in organic solvents to investigate the effect of acid and base. The recovery of Stomidazolone under acidic conditions was determined using ¹H NMR analysis. Please see our revised Fig. S3 and Document S1 for details and the actual data.

To address the stability of Stomidazolone in our experimental aqueous conditions, we incubated Stomidazolone in 1/2 MS media (pH 5.7), which we use for our seedling bioassays, and neutral phosphate buffer (pH 7.0), a generic condition for biochemical assays. The solutions were incubated at room temperature for 48 hours. Subsequently we performed liquid chromatography-mass spectrometry. As shown in Revised Fig. S4, based on our HPLC chromatograms, we did not detect the degradation of Stomidazolone within this time frame. Thus, we can safely say that Stomidazolone is quite stable in the biologically relevant conditions.

Moreover, we have identified the major degradation product of Stomidazolone, which we have designated as Izx103. Subsequent bioassays on Izx103 confirmed that it does not

show any biological activity on stomatal development (see Revised Fig. S3), thus ruling out the possibility that the inhibition of stomatal differentiation is attributable to the degradation product rather than Stomidazolone itself. The yield of Izx103 was determined through isolation. Please see the schematics below (presented as Fig. S3). These new findings are described in detail in our Revised Results section (see Page 6 "Stability of Stomidazolone and identification of inactive Stomidazolone analogs"). Thank you.

The authors should also provide a better ^{13}C NMR spectrum of the compound in the SI, the current copy features rather weak signal intensities and some signals can only be suspected.

We re-synthesized Stomidazolone and substituted with a better ^{13}C NMR spectrum of Stomidazolone in our revised Document S1. Thank you.

Finally, it should be mentioned in the MS right from the beginning that the compound is a racemic mixture (see also point 2 for this).

We sincerely appreciate Reviewer 2 for raising an important point. Please see our detailed response below.

2) A drawback of the study is a lack of suitable controls that unambiguously show that the observed effects are caused by Stomidazolone-mediated modulation of MUTE function. The authors use SIM1 during the plant assays which is however not an optimal control as it is not inactive but displays an alternative activity in stomatal development (reported in a previous paper by the authors). They also do not employ any "inactive" control in the biochemical assays. In my view it will be essential to show that phenotypically inactive compounds are also inactive in the biochemical assays, while active compounds are in contrast active in both assay types. An almost optimal control for these phenotypic and biochemical assays in my view are the two enantiomers of Stomidazolone. As the authors claim that a discrete molecular interaction is the molecular basis of the phenotype and as Stomidazolone is a very small molecule with a rather small and discrete 3D surface, both enantiomers should display different activities in the phenotypic screen as well as in the biochemical assays. They are "optimal controls" as both compounds, due to their stereoisomeric nature, display similar "unspecific effects", e.g. plant distribution properties, etc.. Indeed, different stereoisomers are frequently used as controls in

biological and biochemical assays (in which they are sometimes called eutomer and distomer). The authors report that Stomidazolone is accessible by a simple one-step synthesis in more than 100 mg yield. It should therefore be feasible to perform a subsequent enantiomeric separation of the currently racemic mixture to obtain some mg material of the separated enantiomers and to then test them in parallel in the most important plant and biochemical assays (there is **no need to redo** all plant assays but they should show that one enantiomer induces the desired phenotypic effect while the other is inactive). I would also recommend to have a relook at the binding pose analysis underlying Fig. 6b as two enantiomers should not display equal binding poses.

Again, we thank Reviewer 2 for the thoughtful comments. We recognize that Reviewer 2 may have suggested an enantiometric separation to fulfill the lack of 'true' inactive controls that possess similar structural backbones to Stomidazolone but are inactive in both biological and biochemical contexts. We explored various separation conditions using HPLC equipped with chiral column but were unable to achieve chiral separation. Subsequently, we outsourced the separation to DAICEL (<https://www.daicel.com/en/>), a globally renowned company specializing in chiral separation, but this attempt was also unsuccessful. Based on these results, we determined that separating the enantiomers is not feasible.

Following this, we decided to go back to the foundation of chemical biology. We synthesized analogs of Stomidazolone and investigated their bioactivity on stomatal development, *in silico* docking modeling, and protein binding kinetics. Specifically, we synthesized AYSJ1059 and 1061 from methyl-substituted oxazole and bromo-substituted oxazole using the copper-catalyzed conditions (see synthesis schemes below; also presented in our revised Document S1).

Our seedling bioassays show that these two Stomidazolone analogs do not influence stomatal development, thus they are biologically inactive (see Revised Fig. 2). We subsequently tested *in vitro* whether these Stomidazolone analogs bind to MUTE protein, and whether they interfere with SCR-MUTE heterodimers. As shown in Revised Figs. S6c, d and S8d, e, and Revised Tables S2 and S3, the binding affinity of SCR-MUTE heterodimers are not affected by the presence or absence of AYSJ1059 or AYSJ1061, suggesting that they do not interfere with heterodimer formation. Furthermore, we did not detect any binding of these two analogs to the MUTE_ACTL domain with ITC, and

substantially reduced binding with BLI (see Revised Fig. S7I, m, Tables S2 and S3) . Consistent with these experimental data, molecular docking simulation suggests non-binding/weak binding of AYSJ1059 and 1061 to MUTE ACTL domain (see Revised Fig. S9).

In summary, during the revision period, we successfully synthesized and identified two biologically inactive analogs of Stomidazolone, demonstrating their reduction/ineffectiveness in binding to MUTE to interfere with SCRM-MUTE heterodimers. We are convinced that all these new data fulfill the requirement for proper controls for publication. Two researchers, Kota Hashimoto and Zixuan Li, are now included as co-authors; they performed chemical synthesis and NMR analysis of the inactive Stomidazolone analogs. Thank you so much again for the insightful suggestions.

3) I am also a little bit worried about the raw data of the ITC measurements. I appreciate that the authors have done these measurements as they confirm the molecular model underlying the mode-of-action of the compounds. The ITC measurements are also much more direct and meaningful than BLI measurements which may be influenced by many unspecific events and I acknowledge that the authors have done a lot of work here. However, an ITC Wiseman plot should be sigmoidal (and not almost linear as the plots depicted in Supp Fig. S3). Also important, an upper plateau level should be reached that indicates saturation of the binding site and thus specific binding. Currently, these plots do not look very convincing and seem to indicate unspecific effects, e.g. protein aggregation, etc.? Are the expressed proteins adequately folded? If yes, what model can explain the shape of these curves be explained (and allow data extraction from them)?

We appreciate Reviewer 2's comment. We have provided ITC and BLI measurements as each technique possesses pros and cons. Overall, both our ITC and BLI data are consistent and support the mode of action of Stomidazolone. During the revision, we re-purified the proteins and re-performed ITC assays. Our new ITC raw and analyzed data are presented in Revised Figs. S6, S7, Tables S2 and S3. Essentially, our new ITC measurements support our original values, with slight changes in the exact Kd values. We updated the Kd values in the Tables and in the main text (page 10, 2nd paragraph).

Besides these main points, some further minor points:

4) Although the MS is overall well written and logically structured, there seem to be some expressions/wordings that are perhaps better rephrased. Here only some examples – line 47 “vales” should read as “valves”?;

Yes. Typo corrected.

We also went through wording/expression throughout the manuscript to improve the writing.

line 140: “we took a chemical-genetic approach” – not sure, if this is a “classical” chemical genetic approach which is normally used to describe a screening approach with chemical libraries; the approach here is perhaps better described as “mechanistic studies with mutants”;

Following your advice, we revised the sentence to "mechanistic studies with mutants..."

line 114: “acute angular shape”?; line 79: “Stomata are not the exception” should read as “Stomata are no exception”?; “To expand a chemical space to control stomatal function...” – this sounds somehow strange. There are more examples, please have a critical relook at the text.

We went over our manuscript and elaborated the writing styles throughout. Thank you again for the comments.

5) As stated above, I am no expert on the plant assays and I assume that they have been done with the necessary care and accurateness. I am only wondering on the statistical analysis (which again I assume is overall correct). I understand that the statistically difference group assignments (e.g. a, b, c in Fig. 3) are the outcome of the applied statistical analysis. However, should these assignments not also reflect somehow “obvious” statistical differences? For example, in Fig. 3d, all mock treated data points belong to group a and Stomidazolone-treated samples belong to group b (which makes sense). In Fig. 3f however, the mock treated samples belong to a, but also the t- and s-Stomadizolone samples belong to a, only the m-sample is b (which I find difficult to grasp by looking on the data distribution in the figure). In some measurements, samples even belong to “ab” groups? Again, I think that the authors know what they are doing here but it might be worth to add some explanatory words on this data analysis in the SI for interested readers (like me).

We sincerely thank Reviewer 2 for raising this point (as some of us also felt that these statistical designation of 'a', 'b' etc are somewhat confusing). Originally, the statistical analyses were done separately for different cell types. As such, 'a' for stomata and 'a' for meristemoids are different groups. To be clear, in the revised figures (except Figs. 1e, S2b, c, and S5c as we believe it is more clear leaving as it is), we provide actual pairwise comparison and p values directly indicated in the boxplots. Thank you.

6) Please ensure that all figures are uniformly labeled, e.g. in Fig. 2c,d – y axis is labeled as “/mm²” while in Fig. 3, the y-axis is labeled as “/ 1 mm²”. I also find it difficult to distinguish the colors (green, cyan) in the box plots. Black dots indicate outliers? The outlier selection is based on which calculation?

In response to Reviewer 2's comments, we have refined graph labeling. We adjusted the green color with yellow-green tint that is easier to distinguish from cyan tones.

Reviewer 2 is correct that the black dots in our boxplots indicate outliers. Throughout the manuscript, we use R ggplot2 boxplot function to visualize quantitative data of numbers of stomata, meristemoids, and epidermal cells. In the ggplot2 boxplot, if the data point is greater than $Q3 + 1.5 \cdot IQR$ (above the third quartile greater than 1.5 times of the interquartile range) or less than $Q1 - 1.5 \cdot IQR$, that data point is classified as outlier. In the Materials and Methods of our original manuscript, we have a section “Quantification and statistical analysis”. In our revision, we now include the specific definition of the outliers and also mention their colors (e.g. black dots). Thank you.

REVIEWER COMMENTS

Reviewer #1 (Remarks to the Author):

The authors have invested significant effort in conducting new experiments and analyses, effectively addressing most of my previous comments. However, I have a few minor suggestions that could further improve their work.

Specifically, my comments relate to questions 1 and 2, focusing on SCRM2 from the initial review. The authors suggest that Stomidazolone's observed effects on the *scrm* mutant may stem from its impact on the SCRM2-SPCH heterodimer. However, due to challenges in producing sufficient SCRM2 protein for in vitro ITC analysis, could alternative methods like Y2H and rBiFC be explored to address this issue? Similarly, for question 2, could these methods be employed to elucidate how Stomidazolone influences SCRM2-MUTE heterodimerization, thereby explaining the responses seen in mutants *scrmD_S343* and *scrm-D_S423* to Stomidazolone? Lastly, what are the effects of Stomidazolone on the *scrm scrm2* double mutant?

Reviewer #2 (Remarks to the Author):

The authors have made substantial efforts to address all my concerns. They have adequately addressed all issues and I am therefore happy to recommend publication in its present form.

We thank the Reviewers for their thoughtful comments. We have addressed all the remaining points raised by Reviewer 1. Our point-by-point response is in boldface.

Reviewer #1 (Remarks to the Author):

The authors have invested significant effort in conducting new experiments and analyses, effectively addressing most of my previous comments. However, I have a few minor suggestions that could further improve their work.

Specifically, my comments relate to questions 1 and 2, focusing on SCRM2 from the initial review. The authors suggest that Stomidazolone's observed effects on the *scrm* mutant may stem from its impact on the SCRM2-SPCH heterodimer. However, due to challenges in producing sufficient SCRM2 protein for in vitro ITC analysis, could alternative methods like Y2H and rBiFC be explored to address this issue? Similarly, for question 2, could these methods be employed to elucidate how Stomidazolone influences SCRM2-MUTE heterodimerization, thereby explaining the responses seen in mutants *scrmD_S343* and *scrm-D_S423* to Stomidazolone?

We deeply appreciate Reviewer 1's persistent interest in the role of SCRM2. As suggested, we performed rBiFC assays using full-length SCRM2. SCRM2 forms heterodimers with SPCH, MUTE, and FAMA (see Revised Fig. S6). Most importantly, we observed that Stomidazolone application disrupts SCRM2-SPCH heterodimerization, whereas it has minimal effects on SCRM-SPCH heterodimerization (see Revised Fig. S6). These new results fully explain our original observation that Stomidazolone can confer weak *spch*-like phenotype when applied on *scrm* mutant.

For Reviewer 1's question 2, we further performed Stomidazolone treatment to test SCRM2-MUTE heterodimerization. As expected Stomidazolone disrupts SCRM2-MUTE heterodimerization in rBiFC. The result fully explains the responses seen in *scrm-D_S343* and *scrm-D_S423* (see Revised Fig. 5q, r). Lastly, we additionally show that Stomidazolone also disrupts SCRM2-FAMA heterodimerization (but as we mentioned previously, one would not appreciate *in vivo* effects of Stomidazolone on FAMA, since stomatal precursor cells arrest before the FAMA step).

In summary, our new data address all the remaining questions, resolve the phenotypic consequence of Stomidazolone on *scrm* mutant, and strengthen our original conclusion of how Stomidazolone impacts stomatal bHLH heterodimerization. We would like to re-emphasize that the intricate role of SCRM2 becomes clear in the absence of SCRM. In the presence of SCRM, Stomidazolone preferentially interferes with the MUTE step.

We revised our manuscript text accordingly (see Page 8, L238-244; Page 10, L295-306; Page 14, 3rd paragraph). Plant materials, Oligo DNA primers used for cloning the SCRM2 construct as well as plasmid information is added to Methods, Table S5, and Table S6, respectively. We sincerely appreciate Reviewer 1's suggestions that strengthened our manuscript.

Lastly, what are the effects of Stomidazolone on the *scrm scrm2* double mutant?

As we reported previously (Kanaoka et al. 2008), *scrm scrm2* double mutant produces an epidermis solely composed of pavement cells, just like *spch*. As expected, Stomidazolone has no effects on *scrm scrm2* epidermis: the cotyledon epidermis has

pavement cells only, regardless of mock or Stomidazolone application. We provide the new data as revised Fig. 5g and h.

To make the story complete, we additionally performed Stomidazolone application to *scrm2-1* single mutant. *scrm2-1* does not exhibit any discernible phenotype (looks just like wild type: Kanaoka et al. 2008). As expected, *scrm2-1* seedlings responded just like wild-type seedlings to Stomidazolone application (see Fig. 5e and f of our revised manuscript). We also revised the Results section to include these observations (see Page 8, L238-244; Page 14, 3rd paragraph). Thank you.

Reviewer #2 (Remarks to the Author):

The authors have made substantial efforts to address all my concerns. They have adequately addressed all issues and I am therefore happy to recommend publication in its present form.

Thank you, Reviewer 2, for recommending the publication of our work!

REVIEWERS' COMMENTS

Reviewer #1 (Remarks to the Author):

The revised manuscript has addressed all of my concerns, and I have no further questions.